
# 1  Sea Ice Changes in the Southwest Pacific Sector of the Southern Ocean During the Last 140,000 Years

Jacob Jones[1], Karen E Kohfeld[1,2], Helen Bostock[3, 4], Xavier Crosta[5], Melanie Liston[6], Gavin Dunbar[6], Zanna Chase[7], Amy Leventer[8]
Harris Anderson[7], Geraldine Jacobsen[9]
[1] School of Resource and Environmental Management, Simon Fraser University, Burnaby, Canada
[2] School of Environmental Science, Simon Fraser University, Burnaby, Canada
[3] School of Earth and Environmental Sciences, The University of Queensland, Brisbane, Australia
[4] National Institute of Water and Atmospheric Research (NIWA), Wellington, New Zealand
[5] Université de Bordeaux, CNRS, EPHE, UMR 5805 EPOC, Pessac, France
[6] Antarctic Research Centre, Victoria University of Wellington, Wellington, New Zealand
[7] Institute of Marine and Antarctic Studies, University of Tasmania, Hobart, Australia
[8] Geology Department, Colgate University, Hamilton, NY, USA
[9] Australian Nuclear Science and Technology (ANSTO), Lucas Heights, New South Wales, Australia
*Correspondence to*: Jacob Jones (jacob_jones@sfu.ca)

## 18  Abstract

Sea ice expansion in the Southern Ocean is believed to have contributed to glacial-interglacial
atmospheric $CO_2$ variability by inhibiting air-sea gas exchange and influencing the ocean's
meridional overturning circulation. However, limited data on past sea ice coverage over the last
140 ka (a complete glacial cycle) have hindered our ability to link sea ice expansion to oceanic
processes that affect atmospheric $CO_2$ concentration. Assessments of past sea ice coverage
using diatom assemblages have primarily focused on the Last Glacial Maximum (~21 ka) to
Holocene, with few quantitative reconstructions extending to the onset of glacial Termination II
(~135 ka). Here we provide new estimates of winter sea ice concentrations (wSIC) and summer
sea surface temperatures (sSSTs) for a full glacial-interglacial cycle from the southwestern
Pacific sector of the Southern Ocean using fossil diatom assemblages from deep-sea core
TAN1302-96 (59.09°S, 157.05°E, water depth 3099 m). We find that winter sea ice was
consolidated over the core site during the latter part of the penultimate glaciation, Marine
Isotope Stage (MIS) 6 (from at least 140 to 134 ka), when sSSTs were between ~1 and 1.5°C.
The winter sea ice edge then retreated rapidly as sSSTs increased during the transition into the
Last Interglacial Period (MIS 5e), reaching ~4.5°C by 125 ka. As the Earth entered the early
glacial stages, sSSTs began to decline around 112 ka, but winter sea ice largely remained absent
until ~65 ka during MIS 4, when it was sporadically present but unconsolidated (<40% wSIC).
WSIC and sSSTs reached their maximum concentration and coolest values by 24.5 ka, just prior
to the Last Glacial Maximum. Winter sea ice remained absent throughout the Holocene, while
SSSTs briefly exceeded modern values, reaching ~5°C by 11.4 ka, before decreasing to ~4°C and
stabilizing. The absence of sea ice coverage over the core site during the early glacial period
suggests that sea ice may not have been a major contributor to $CO_2$ drawdown at this time.
During MIS 5d, we observe a weakening of meridional SST gradients between 42° to 59°S
throughout the region, which may have contributed to early reductions in atmospheric $CO_2$
concentrations through its impact on air-sea gas exchange. Sea ice expansion during MIS 4,
however, coincides with observed reductions in Antarctic Intermediate Water production and



subduction, suggesting that sea ice may have influenced intermediate ocean circulation
changes.

## 1.0 Introduction

Antarctic sea ice has been suggested to have played a key role in glacial-interglacial
atmospheric $CO_2$ variability (e.g., Stephens & Keeling, 2000; Ferrari et al., 2014; Kohfeld &
Chase, 2017; Stein et al., 2020). Sea ice has been dynamically linked to several processes that
promote deep ocean carbon sequestration, namely by: [1] reducing deep ocean outgassing by
ice-induced 'capping' and surface water stratification (Stephens & Keeling, 2000; Rutgers van
der Loeff et al., 2014), and [2] influencing ocean circulation through water mass formation and
deep-sea stratification, leading to reduced diapycnal mixing and reduced $CO_2$ exchange
between the surface and deep ocean (Toggweiler, 1999; Bouttes et al., 2010; Ferrari et al.,
2014). Numerical modelling studies have shown that sea ice-induced capping, stratification, and
reduced vertical mixing may be able to account for a significant portion of the total $CO_2$
variability on glacial-interglacial timescales (between 40-80 ppm) (Stephens & Keeling, 2000;
Galbraith & de Lavergne, 2018; Marzocchi & Jansen, 2019; Stein et al., 2020). However, debate
continues surrounding the timing and magnitude of sea ice impacts on glacial-scale carbon
sequestration (e.g., Morales Maquede & Rahmstorf, 2002; Archer et al., 2003; Sun &
Matsumoto, 2010; Kohfeld & Chase, 2017).
Past Antarctic sea ice coverage has been estimated primarily through diatom-based
reconstructions, with most work focusing on the Last Glacial Maximum (LGM), specifically the
EPILOG timeslice as outlined in Mix et al. (2001), corresponding to 23 to 19 thousand years ago
(ka). During the LGM, these reconstructions suggest that winter sea ice expanded by 7-10°
latitude (depending on the sector of the Southern Ocean), which corresponds to an
approximate doubling of total winter sea ice coverage compared to modern observations
(Gersonde et al., 2005; Benz et al., 2016). Currently, only a handful of studies provide
quantitative sea ice coverage estimates back to the penultimate glaciation, Marine Isotope
Stage (MIS) 6 (~194 to 135 ka) (Gersonde & Zielinksi, 2000; Crosta et al., 2004; Schneider-Mor
et al., 2012; Esper & Gersonde 2014; Ghadi et al. 2020). These studies primarily cover the
Atlantic sector, with only one published sea ice record from each of the Indian (SK200-33 from



Ghadi et al., 2020), eastern Pacific (PS58/271-1 from Esper & Gersonde, 2014), and
southwestern Pacific sectors (SO136-111 from Crosta et al., 2004). These glacial-interglacial sea
ice records show heterogeneity between sectors in both timing and coverage. While the
Antarctic Zone (AZ) in the Atlantic sector experienced early sea ice advance corresponding to
MIS 5d cooling (i.e., 115 to 105 ka) (Gersonde & Zielinksi, 2000; Bianchi & Gersonde, 2002;
Esper & Gersonde, 2014), the Indian and Pacific sector cores in the AZ show only minor sea ice
advances during this time (Crosta et al., 2004; Ghadi et al., 2020). The lack of spatial and
temporal resolution has resulted in significant uncertainty in our ability to evaluate the timing
and magnitude of sea ice change during a full glacial cycle across the Southern Ocean, and to
link sea ice to glacial-interglacial $CO_2$ variability.

This paper provides new winter sea ice concentration (wSIC) and summer sea surface

temperature (sSST) estimates for the southwestern Pacific sector of the Southern Ocean over
the last 140 ka. SSSTs and wSIC are estimated by applying the Modern Analog Technique (MAT)
to fossil diatom assemblages from sediment core TAN1302-96 (59.09°S, 157.05°E, water depth
3099 m). We place this record within the context of sea ice and sSST changes from the region
using previously published records from SO136-111 (56.66°S, 160.23°E, water depth 3912 m),
which has recalculated wSIC and sSST estimates presented in this study, and nearby marine
core E27-23 (59.61°S, 155.23°E; water depth 3182 m) (Ferry et al., 2015). Using these records,
we compare the timing of sea ice expansion to early glacial-interglacial $CO_2$ variability to test
the hypothesis that the initial $CO_2$ drawdown (~115 to 100 ka) resulted from reduced air-sea
gas exchange in response to sea ice capping and surface water stratification. We then consider
alternative oceanic drivers of early atmospheric $CO_2$ variability, and place our sSSTs estimates
within the context of other studies to examine how regional cooling and a weakening in
meridional SST gradients might affect air-sea disequilibrium and early $CO_2$ drawdown (Khatiwala
et al., 2019). Finally, we compare our wSIC estimates with regional reconstructions of Antarctic
Intermediate Water (AAIW) production and subduction variability using previously published
carbon isotope analyses on benthic foraminifera from intermediate to deep-water depths in the
southwest Pacific sector of the Southern Ocean to test the hypothesis that sea ice expansion is
dynamically linked to AAIW production and variability (Ronge et al., 2015).





## 2.0 Methods


### 2.1 Study Site & Age Determination


We reconstruct diatom-based wSIC and sSST using marine sediment core TAN1302-96
(59.09°S, 157.05°E, water depth 3099 m) (Figure 1). The 364 cm core was collected in March
2013 using a gravity corer during the return of the *RV Tangaroa* from the Mertz Polynya in
Eastern Antarctica (Williams et al., 2013). The core is situated in the western Pacific sector of
the Southern Ocean, on the southwestern side of the Macquarie Ridge, approximately 3-4°
south of the average position of the Polar Front (PF) at 157°E (Sokolov & Rintoul, 2009).

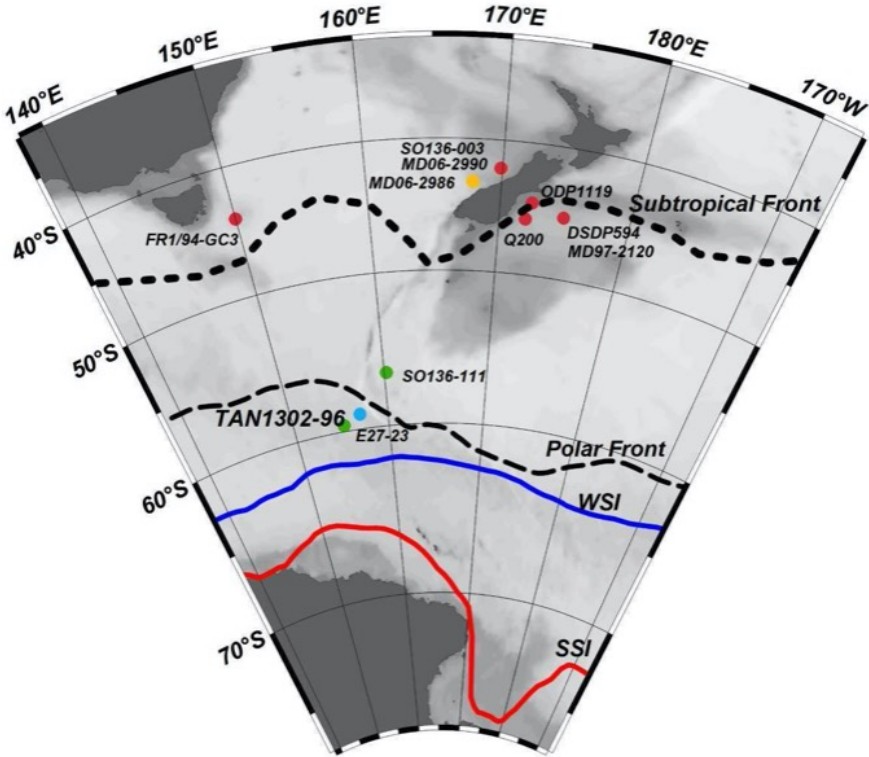


**Figure 1:** Map of the southwestern Pacific sector of the Southern Ocean including the study
site, TAN1302-96 (blue circle), and additional cores providing supporting information on sea ice
extent, SO136-111 and E27-23 (green circles), SST reconstructions (red circles), and $\delta^{13}$C of
benthic foraminifera (yellow circles). Metadata for these cores are provided in Table A1.
Dashed lines show the average location of the Subtropical and Polar Fronts (Smith et al., 2013;



Bostock et al., 2015), red and blue lines show approximate positions of summer and winter sea
ice extents, respectively (Reynolds et al., 2002; 2007).

The age model for TAN1302-96 (Figures 2 and 3) was based on a combination of

radiocarbon dating of mixed foraminiferal assemblages, and stable oxygen isotope stratigraphy
on *Neogloboquadrina pachyderma* (180-250 µm). Seven accelerator mass spectrometry (AMS)
[14]C samples were collected (Table A1 in Appendix A) and consisted of mixed assemblages of
planktonic foraminifera (*N. pachyderma* and *Globigerina bulloides,* >250 µm). Three of the
seven radiocarbon samples (NZA 57105, 57109, and 61429) were previously published in
Prebble et al., (2017), and four additional samples (OZX 517-520) were added to improve the
dating reliability (Table A1 in Appendix A). OZX 519 and OZX 520 produced dates that were not
distinguishable from background (>57.5 ka) and were subsequently excluded from the age
model. The TAN1302-96 oxygen isotopes were run at the National Institute of Water and
Atmospheric Research (NIWA) using the Kiel IV individual acid-on-sample device and analysed
using Finnigan MAT 252 Mass Spectrometer. The precision is ±0.07% for $\delta^{18}$O and ±0.05% for
$\delta^{13}$C.

The age model was constructed using the 'Undatable' MATLAB software by

bootstrapping at 10% and using an x-factor of 0.1 (Lougheed & Obrochta, 2019), which scales
Gaussian distributions of sediment accumulation uncertainty (Table A2 in Appendix A). Below
100 cm, six tie points were selected at positions of maximum change in $\delta^{18}$O and were
correlated to the LR04 benthic stack (Lisiecki & Raymo, 2005) (Fig 2; Table A2 in Appendix A).
We used a conservative marine reservoir age (MRA) for radiocarbon calibration of 1000 +/- 50
years, in line with regional estimates in Paterne et al. (2019) and modelled estimates by Butzin
et al. (2017; 2020). The age model shows that TAN1302-96 extends to at least 140 ka, capturing
a full glacial-interglacial cycle. Linear sedimentation rates (LSR) in TAN 1302-96 were observed
to be higher during interglacial periods, averaging 3.37cm/ka, compared to glacial periods,
averaging 2.74cm/ka. It is worth noting that there can be significant MRA variability over time
due changes in ocean ventilation, sea ice coverage, and wind strength, specifically in the polar



high latitudes (Heaton et al., 2020), and as a result, caution should be taken when interpreting
the precision of radiocarbon dates.

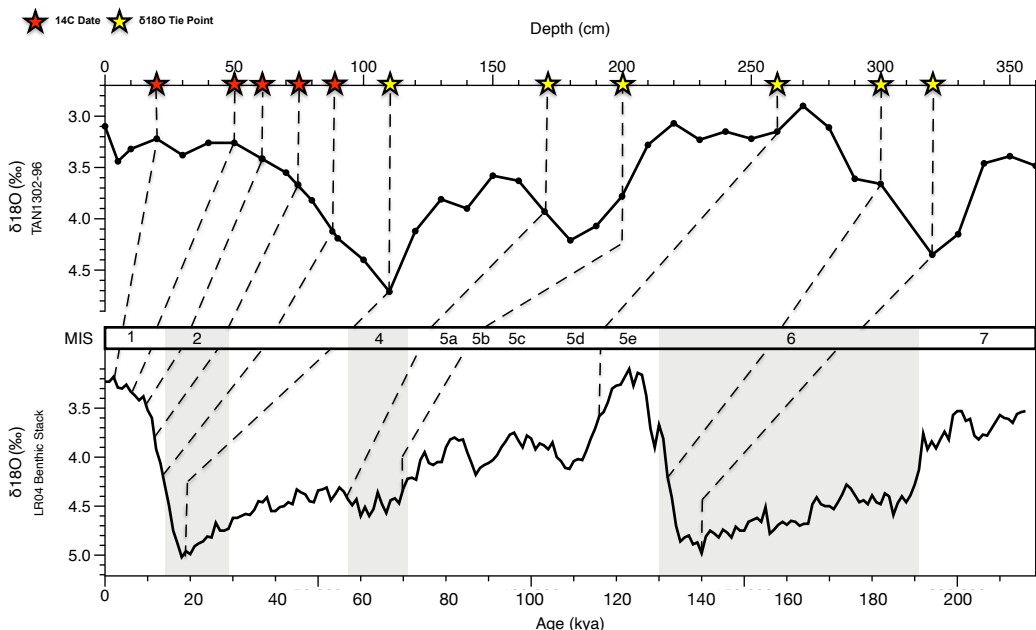

**Figure 2:** Age model of TAN1302-96. Red stars indicate the depth of AMS [14]C samples, and
yellow stars indicate tie points between the TAN1302-96 oxygen isotope stratigraphy and the
LR04 benthic stack (Lisiecki & Raymo, 2005). Two radiocarbon dates, OZX 519 & 520, were not
included in the age model as they produced dates that were NDFB (not distinguishable from
background).

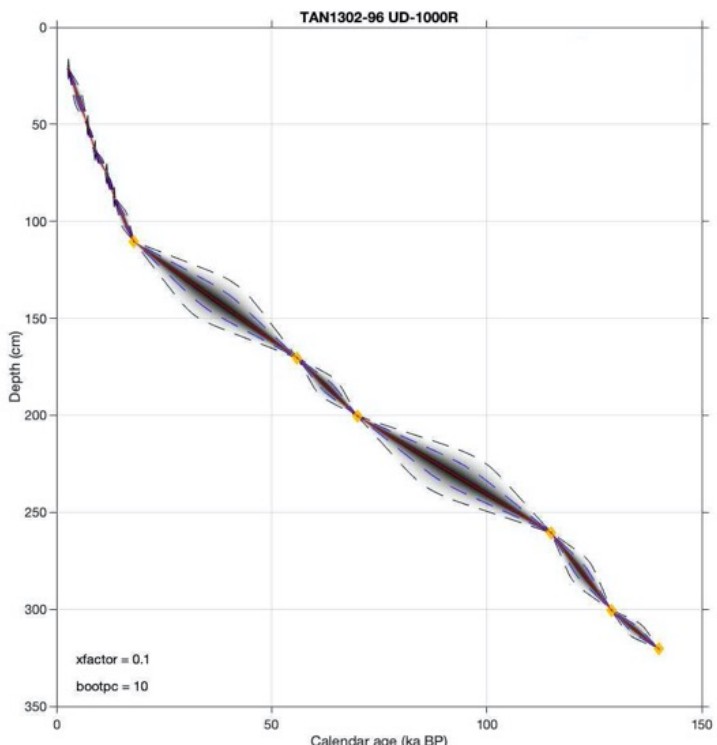

**Figure 3:** Age model of TAN1302-96. Tie points are depicted as yellow dots and grey shading represents associated uncertainty between tie points. The age model used a marine reservoir calibration of 1000 +/- 50 years.

## 2.2 Diatom Analysis

TAN1302-96 was sampled every 3-4 cm throughout the core except between 130-180 cm, where samples were collected every 10 cm due to limited availability of sample materials (Table A3 in Appendix A). Diatom slide preparation followed two procedures. The first approach approximated the methods outlined in Renberg (1990), while the second followed the protocol outlined in Warnock & Scherer (2014). To ensure there were no biases between preparation techniques, results from each technique were first visually compared followed by a comparison of sample means (see Figure B1 in Appendix B). No biases in the data were observed between methods.

The first procedure was conducted at Victoria University of Wellington and Simon Fraser
University on samples every 10 cm throughout the core. Sediment samples contained high
concentrations of diatoms with little carbonaceous or terrigenous materials, so no dissolving
aids were used. Instead, approximately 50 mg of sediment was weighed, placed into a 50 ml
centrifuge tube, and topped up with 40 ml of deionized water. Samples were then manually
shaken to disaggregate sediment, followed by a 10-second mechanical stir using a vortex
machine. Samples were then left to settle for 25 seconds. 0.25 mL of the solution was then
pipetted onto a microscope slide from a consistent depth, where it was left to dry overnight.
Once the sample had dried, coverslips were permanently mounted to the slide using Permount,
a high refractive index mountant. Slides were redone if they contained too many diatoms and
identification was not possible, or if they contained too few diatoms (generally <40 specimens
per transect). Sediment sample weight was adjusted to achieve the desired dilution.
The second procedure was conducted at Colgate University on samples every 3-4 cm
throughout the core. Oven-dried samples were placed into a 20 ml vial with 1-2 ml of 10% $H_2O_2$
and left to react for up to several days, followed by a brief (2-3 second) ultrasonic bath to
disaggregate samples. The diatom solution was then added into a settling chamber, where
microscope coverslips were placed on stages to collect settling diatoms. The chamber was
gradually emptied through an attached spigot, and samples were evaporated overnight. Cover
slips were permanently mounted onto the slides with Norland Optical Adhesive #61, a
mounting medium with a high refractive index.
Diatom identification was conducted at Simon Fraser University using a Leica Leitz
DMBRE light microscope using standard microscopy techniques. Following transverses, a
minimum of 300 individual diatoms were identified at 1000x magnification from each sample
throughout the core. Individuals were counted towards the total only if they represented at
least one-half of the specimen so that fragmented diatoms were not counted twice.
Identification was conducted to the highest taxonomic level possible, either to the species or
species-group level. Taxonomic identification was conducted using numerous identification
materials, including (but not limited to): Fenner et al. (1976); Fryxell & Hasle (1976; 1980);
Johansen & Fryxell (1985); Hasle & Syversten (1997); Cefarelli et al. (2010); and Wilks & Armand





(2017). Diatom species that have similar environmental preferences were grouped together as
outlined in Crosta et al. (2004). Three main taxonomic groups were established, and their
relative abundances were calculated by dividing the number of identified specimens of a
particular species by the total number of identified diatoms from the sample. The following
main taxonomic groups were used (Table 1):

**[1]** Sea Ice Group: representing diatoms that thrive in or near the sea ice margin in SSTs
generally ranging from -1 to 1 °C.
**[2]** Permanent Open Ocean Zone (POOZ): representing diatoms that thrive in open
ocean conditions, with SSTs generally ranging from ~2 to 10 °C.
**[3]** Sub-Antarctic Zone (SAZ): representing diatoms that thrive in warmer sub-Antarctic
waters, with SSTs generally ranging from 11 to 14 °C.

**Table 1**: Species comprising each of the diatom taxonomic groups (updated from Crosta et al.,
2004).

| Sea Ice Group | POOZ Group | SAZ Group |
|---|---|---|
| *Actinocyclus actinochilus* | *Fragilariopsis kerguelensis* | *Azpeitia tabularis* |
| *Fragilariopsis curta* | *Fragilariopsis rhombica* | *Hemidiscus cuneiformis* |
| *Fragilariopsis cylindrus* | *Fragilariopsis separanda* | *Thalassionema nitzschioides var. lanceolata* |
| *Fragilariopsis obliquecostata* | *Rhizosolenia polydactyla* var. *polydactyla* | *Thalassiosira eccentrica* |
| *Fragilariopsis ritscheri* | *Thalassionema nitzschioides* (form 1) | *Thalassiosira oestrupii* gp. |
| *Fragilariopsis sublinearis* | *Thalassiosira gracilis* gp. | |
| | *Thalassiosira lentiginosa* | |
| | *Thalassiosira oliverana* | |
| | *Thalassiothrix* sp. | |
| | *Trichotoxon reinboldii* | |


**2.3 Modern Analog Technique**
Past wSIC and sSSTs (January to March) were estimated for TAN1302-96 and
recalculated for SO136-111 by applying the Modern Analog Technique (MAT) to the fossil
diatom assemblages, as outlined in Crosta et al. (1998; 2020). The MAT reference database
used for this analysis is comprised of 249 modern core top samples (analogs) located primarily
in the Atlantic and Indian sectors from ~40°S to the Antarctic coast. The age of the core tops





included in the reference database have been assessed through radiocarbon and/or isotope
stratigraphy when possible. Core tops were visually evaluated for selective diatom dissolution,
so it is believed that sub-modern assemblages contain well-preserved and unbiased specimens.
Modern summer SSTs and wSIC were interpolated from the reference core locations using a
1°x1° grid from the World Ocean Atlas (Locarnini et al., 2013) through the Ocean Data View
(Schlitzer, 2005). The MAT was applied using the "bioindic" package (Guiot & de Vernal, 2011)
through the R-platform. Fossil diatom assemblages were compared to the modern analogs
using 33 species or species-groups to identify the five most similar modern analogs using both
the LOG and CHORD distance. The reconstructed sSST and wSIC are the distance-weighted
mean of the climate values associated with the selected modern analog (Guiot et al., 1993;
Ghadi et al., 2020). Both MAT approaches produce an $R^2$ value of 0.96 and a root mean square
error of prediction (RMSEP) of ~1 °C for summer SST, and an $R^2$ of 0.93 and a RMSEP of 10% for
wSIC (Ghadi et al. 2020). As outlined in Ferry et al., (2015), we consider <15% wSIC to represent
an absence of winter sea ice, 15-40% wSIC as present but unconsolidated, and >40% to
represent consolidated winter sea ice.




## 3.0 Results

### 3.1 Diatom Assemblage Results

Fifty-one different species or species groups were identified, of which 33 were used in
the transfer function. Polar Open Ocean Zone (POOZ) diatoms made up the largest proportion
of diatoms identified, representing between 72-91% of the assemblage (Figure 4), with higher
values observed during warmer interstadial periods of MIS 1, 3, and 5. Sea ice diatoms made up
the second most abundant group, representing between 0.5-7.5% of the assemblage, with
higher values observed during cooler stadial periods (MIS 2, 4, and 6). The Sub-Antarctic Zone
group had relatively low abundances, with higher SAZ values occurring during warmer
interstadial periods.

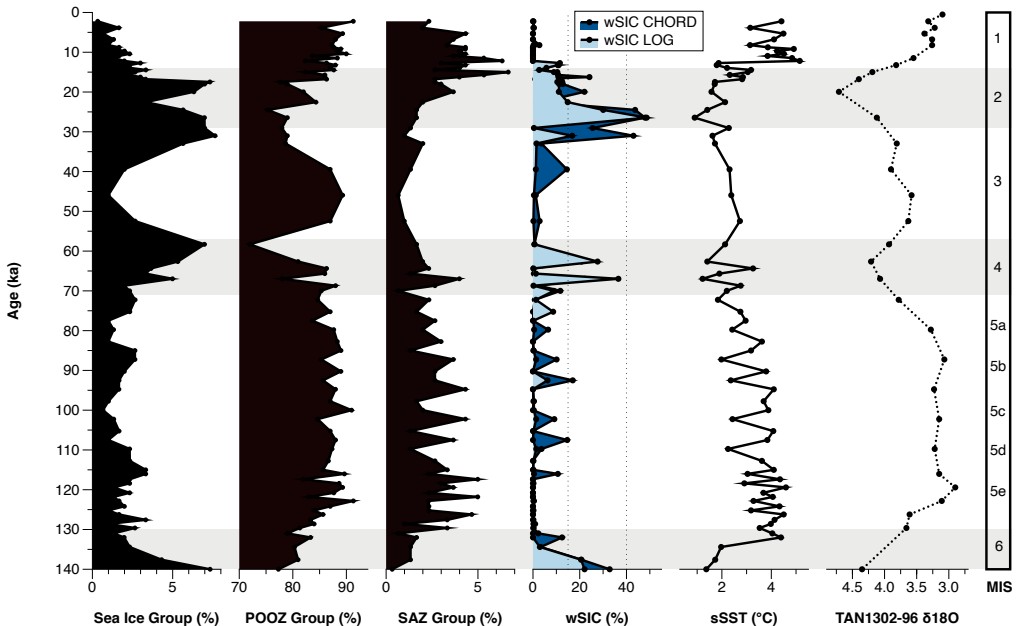


**Figure 4:** Diatom assemblages results from TAN1302-96 separated into % contribution from
each taxonomic group (Sea ice Group, POOZ, & SAZ; see Table 1) over a full glacial-interglacial
cycle. Using the Modern Analog Technique (MAT), winter sea ice concentration (wSIC) and
summer sea surface temperature (sSST) were estimated and compared against the $\delta^{18}$O
signature of TAN1302-96.





### 3.2 TAN1302-96 Summer SST and wSIC Estimates

Estimates of sSST and wSIC from both LOG and CHORD MAT outputs produced similar

results (Figure 4). During Termination II, sSSTs began to rise from ~1°C at 140 ka (MIS 6) to 4.5°C
at 130 ka (MIS 5e/6 boundary). This warming corresponded with a decrease in wSIC from 48%
to approximately 0% over the same time periods (Figure 4). Reconstructed sSSTs continued to
rise slightly throughout MIS 5e, reaching a maximum value of 4.6°C at 118 ka, after which they
declined throughout MIS 5. During this period of sSST decline, winter sea ice was largely absent,
punctuated by brief periods during which sea ice was present but unconsolidated (wSIC of
14.7% and 17% at 105 and 90 ka, respectively). During MIS 4 (71 to 57 ka), sSSTs cooled to
between roughly 1°C and 3°C, and sea ice expanded to 36%, such that it was present but
unconsolidated for intervals of a few thousand years. SSSTs increased slightly from 1.5°C at 61
ka (during MIS 4) to ~2.5°C at 50 ka (during MIS 3), followed by a general cooling trend into MIS
2. Sea ice appears to have been largely absent during MIS 3 (57 to 29 ka), although sampling
resolution is low, but increased rapidly to 48% cover during MIS 2 where winter sea ice was
consolidated over the core site. During MIS 2, sSSTs cooled to a minimum of <1°C at 24.5 ka.
After 18 ka, the site rapidly transitioned from cool, ice-covered conditions to warmer, ice-free
winter conditions during the early deglaciation. This warming was interrupted by a brief cooling
around 13.5 ka, following which sSSTs quickly reached their maximum values of ~5°C at 11.4 ka
and remained relatively high throughout the rest of the Holocene. Winter sea ice was not
present during the Holocene. There were no non-analog conditions observed in TAN1302-96
samples.

### 4.0 Discussion

### 4.1 Regional sSST and wSIC Estimates

The new wSIC and sSST estimates from TAN1302-96 and recalculated wSIC estimates

from SO136-111 show a coherent regional pattern (Figure 5). Both cores show relatively high
concentrations of sea ice during MIS 2, 4, and 6, with lower values during MIS 1 and 5.
TAN1302-96 shows slightly higher concentrations during MIS 2 (47%) and 4 (37%) compared
with SO136-111 (35% and 36%, respectively), which can be explained by a more poleward

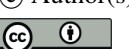

position of TAN1302-96 relative to SO136-111. The estimates between cores differ during MIS
3, with seemingly lower wSIC in TAN1302-96 than in SO136-111, which might result from the
low sampling resolution in TAN1302-96 during this period. Overall, these cores show a highly
similar and coherent history of sea ice over the last 140 ka.

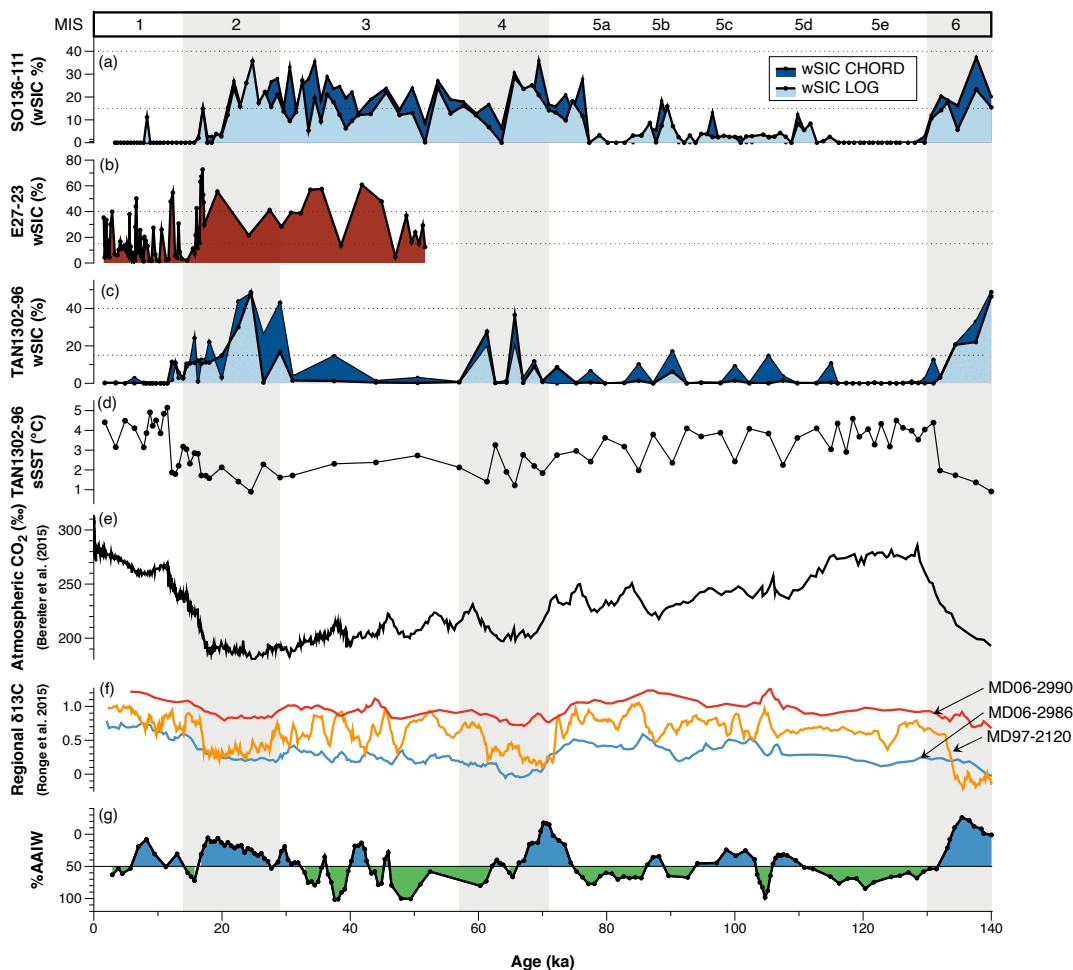

**Figure 5: (a)** wSIC estimates using MAT from SO136-111 (recalculated in this study, see
Appendix D); **(b)** wSIC estimates using GAM from E27-23 (Ferry et al., 2015); **(c)** wSIC estimates
using MAT from TAN1302-96 (this study); **(d)** sSST estimates using MAT from TAN1302-96 (this
study); **(e)** Antarctic atmospheric $CO_2$ concentrations over 140 ka (Bereiter et al., 2015); **(f)** $\delta^{13}C$
data from nearby cores MD06-2990/SO136-003, MD97-2120, and MD06-2986 (Ronge et al.,
2015); **(g)** %Antarctic Intermediate Water (%AAIW) as calculated in Ronge et al. (2015), which





tracks when core MD97-2120 was bathed primarily by AAIW (green) or Upper Circumpolar
Deep Water (UCDW) (blue).

When compared with E27-23 (Figure 5), which is located only ~120 km to the southwest

of TAN1302-96 (Figure 1), the TAN1302-96 core shows lower estimates of wSIC, especially
during MIS 3. During early and mid-MIS 2, both cores show similar wSIC estimates, while later in
MIS 2 (~17 ka), E27-23 reports a maximum wSIC of 72% compared to only 22% at TAN1302-96.
A discrepancy between estimates is also observed during the Holocene, with E27-23 reporting
sea ice estimates of up to nearly 50% during the mid-Holocene (~6 ka), while TAN1302-96
experienced values well below the RMSEP of 10%.

Possible explanations for the observed differences in wSIC estimates include: [1]

differences in laboratory protocols; [2] differences in diatom identification/counting
methodology; [3] differences in statistical applications; [4] selective diatom dissolution; and [5]
differences in the redistribution of sediment by the Antarctic Circumpolar Current (ACC)
between each of the core sites. Of these explanations, we believe that [3] and [5] are the most
likely candidates and are discussed below (for further discussion on [1], [2], and [4], see
Appendix C).

The first possible explanation we identified is through the use of different statistical

applications. Ferry et al. (2015) used a Generalized Additive Model (GAM) to estimate wSIC for
both E27-23 and SO136-111, while we have used the MAT for TAN1302-96 and SO136-111. A
simple comparison of wSIC estimates between the results in Ferry et al. (2015) and our
recalculated wSIC estimates for SO136-111 can provide insights into the magnitude of
estimation differences. Generally speaking, the GAM estimation produced higher wSIC
estimates than the MAT (e.g., ~50% wSIC at 23 ka while the MAT produced ~37% for the same
time period); however, we believe it is unlikely that statistical approaches alone could explain a
larger difference (i.e., 50%) between E27-23 and TAN1302-96.

The second possible explanation is through lateral sediment redistribution and focusing

by the ACC. We estimated sediment focusing for E27-23 using [230]Th data from Bradtmiller et al.
(2009) together with dry bulk density estimated using calcium carbonate content (Froelich,
1991). Both sedimentation rates and focusing factors for the E27-23 are relatively high (~35
cm/ka and max=26, respectively) during the LGM and Holocene, which could influence the
reliability of wSIC and sSST estimation (see Figure B2 in Appendix B). Several peaks in focusing
occurring around 16, 12, and 3 ka appear to closely correspond to periods of peak wSIC (~67%,
~54%, and ~35%, respectively), suggesting a possible link. Lateral redistribution could artificially
increase or decrease relative abundances of some diatom groups, which could lead to over or
under estimations of sea ice coverage. Thorium analysis for TAN1302-96 is beyond the scope of
this study; however, future work could help address this uncertainty.

Although we are unable to identify the specific cause of the differences, we do suggest

using caution when interpreting the exact magnitude of sea ice expansion in this region, and to
consider the results from all cores when drawing conclusions of regional sea ice history.

**4.2 The Role of Sea Ice on Early $CO_2$ Drawdown**

Kohfeld & Chase (2017) hypothesized that the initial drawdown of atmospheric $CO_2$ (~35

ppm) during the glacial inception of MIS 5d (~115 to 100 ka) was primarily driven by sea ice
capping and a corresponding stratification of surface waters, which reduced the $CO_2$ outgassing
of upwelled carbon-rich waters. This hypothesis is supported by several lines of evidence,
including: [1] sea salt sodium (ssNA) archived in Antarctic ice cores, suggesting sea ice
expansion near the Antarctic continent (Wolf et al., 2010); [2] $\delta^{15}$N proxy data from the central
Pacific sector of the Southern Ocean, suggesting increased stratification south of the modern-
day Antarctic Polar Front (Studer et al., 2015); and [3] diatom assemblages in the polar frontal
zone of the Atlantic sector, suggesting a slight cooling and northward expansion of sea ice
during MIS 5d (Bianchi & Gersonde, 2002). Our data address this hypothesis by providing
insights into early sea ice expansion into the polar frontal zone of the western Pacific sector.

Our data show that, in contrast to the Atlantic sector (Bianchi & Gersonde, 2002), there

was no sea ice advance into the modern-day PFZ of the SW Pacific during MIS5d. Neither
TAN1302-96 nor S0136-111 shows evidence of an early glacial increase in wSIC (Figure 5).
Unfortunately, the lack of spatially extensive quantitative records extending back to
Termination II limits our ability to estimate the timing and magnitude of sea ice changes for



regions poleward of 59°S in the southwestern Pacific. We anticipate, however, that an advance
in the sea ice edge, consistent with those outlined in Bianchi & Gersonde (2002), likely would
have reduced local SSTs as the sea ice edge advanced closer to the core site. Indeed, the
TAN1302-96 SST record does show a decrease to ~2°C (observed at 107 ka), which quickly
rebounded to ~4°C by ~105 ka (Figure 5). However, this sSST drop occurred roughly 8 ka after
the initial $CO_2$ reduction, suggesting that the $CO_2$ drawdown event and local sSST reduction may
not be linked. While the lack of sea ice diatoms and no discernable reductions of sSSTs occurred
during MIS 5d at TAN1302-96 or SO136-111, we cannot rule out the possibility that modest sea
ice advances, or a consolidation of pre-existing sea ice, took place south of the core sites.

Given that sea ice was not at its maximum extent during the early glacial, it stands to

reason that any reductions to air-sea gas exchange in response to the hypothetically expanded
sea ice would not have been at its maximum impact either. Thus, it is likely that any effects of
sea ice capping would not have reached their full potential during the early glacial period.
Previous modeling work has suggested that the maximum impact of sea ice expansion on
glacial-interglacial atmospheric $CO_2$ reductions ranged from 5 to 14 ppm (Kohfeld & Ridgwell,
2009). More recent modeling studies are consistent with this range, suggesting a 10-ppm
reduction (Stein et al., 2020), while some studies even suggest a possible increase in
atmospheric $CO_2$ concentrations due to sea ice expansion (Khatiwala et al., 2019). Furthermore,
Stein et al. (2020) suggest that the effects of sea ice capping would have taken place after
changes in deep ocean stratification had occurred and would have contributed to $CO_2$
drawdown later during the mid-glacial period. These model results, when combined with our
data, suggest that even if modest sea ice advances did take place during the early glacial (i.e.,
MIS 5d), their impacts on $CO_2$ variability would likely have been modest, ultimately casting
doubt on the hypothesis that early glacial $CO_2$ reductions of 35 ppm can be linked solely to the
capping and stratification effects of sea ice expansion.

**4.3 Other Potential Contributors to Early Glacial $CO_2$ Variability**

The changes observed in wSIC and sSST from TAN1302-96 suggests that sea ice

expansion was likely not extensive enough early in the glacial cycle for a sea ice capping effect



to be solely responsible for early atmospheric $CO_2$ drawdown. This leaves open the question of
what may have contributed to early drawdown of atmospheric $CO_2$. In terms of the ocean's
role, we highlight three contenders: [1] a potentially non-linear response between sea ice
coverage and $CO_2$ sequestration potential; [2] links between sea ice expansion and early
changes in global ocean overturning, and [3] the impact of cooling on air-sea disequilibrium in
the Southern Ocean.
The first possible explanation considers that not all sea ice has the same capacity to
facilitate or inhibit air-sea gas exchange. We previously suggested that because sea ice was not
at its maximum extent during MIS 5d, the contribution of sea ice on $CO_2$ sequestration would
likely not be at its maximum extent either. However, this assumes that there is a linear
relationship between sea ice coverage and $CO_2$ sequestration potential. We know that different
sea ice properties, such as thickness and temperature, determine overall porosity, with thicker
and colder sea ice being less porous and more effective at reducing air-sea gas exchange
compared to thinner and warmer sea ice (Delille et al., 2014). It is therefore possible that if
modest sea ice advances took place closer to the Antarctic continent (and were therefore not
captured by TAN1302-96), they may have been more effective at reducing $CO_2$ outgassing,
either by experiencing some type of reorganization or consolidation, or through a change in
properties such as temperature or thickness. It is also possible that sea ice coverage over some
regions leads to more effective capping, while in other regions sea ice growth contributes only
to marginal reductions in air-sea gas exchange. This, theoretically, could point to a non-linear
response between sea ice expansion and $CO_2$ sequestration potential, and could link modest
sea ice growth around the Antarctic continent to the ~35 ppm initial $CO_2$ drawdown event.
While this is theoretical and cannot be adequately addressed in this analysis, it is worthy of
deeper consideration.
The second possible explanation involves changes in the global overturning circulation.
Kohfeld & Chase (2017) previously examined the timing of changes in $\delta^{13}C$ of benthic
foraminifera solely from the Atlantic basin and observed that the largest changes in AMOC
coincided with the mid-glacial reductions in atmospheric carbon dioxide changes mentioned
above. Subsequent work of O'Neill et al. (2020) examined whole-ocean changes in $\delta^{13}C$ of



benthic foraminifera and noted that the separation between $\delta^{13}C$ values of abyssal and deep
ocean waters were actually initiated between MIS 5d and MIS 5a (114 to 71 ka). Evidence for
early changes in abyssal circulation have also been detected in Indian Ocean $\delta^{13}C$ records
(Govin et al., 2009), and more recently in Indian Ocean εNd records (Williams et al., 2021),
suggesting that the abyssal ocean may have responded to sea ice changes around the Antarctic
continent early in the glacial cycle. If indications of an early-glacial response in the global ocean
circulation in the Indo-Pacific are correct, these data may also point to an elevated importance
of sea ice near the Antarctic continent in triggering early, deep-ocean overturning changes.

The third possible explanation involves changes in surface ocean temperature gradients

in the Southern Ocean, and how they could influence air-sea gas exchange. Several recent
studies have pointed to the importance of changes to air-sea disequilibrium as a key
contributor to $CO_2$ uptake in the Southern Ocean (Eggleston & Galbraith, 2018; Marzocchi &
Jansen 2019; Khatiwala et al. 2019). Khatiwala et al. (2019) suggested that modelling studies
have traditionally underrepresented (or neglected) the role of air-sea disequilibrium in
amplifying the impact of cooling on potential $CO_2$ sequestration in the mid-high southern
latitudes during glacial periods. They argue that when the full effects of air-sea disequilibrium
are considered, ocean cooling can result in a 44 ppm decrease due to temperature-based
solubility effects alone. They attributed this increased impact of SSTs to a reduction in sea-
surface temperature gradients explicitly in polar mid-latitude regions (roughly between 40° and
60° north and south). If we compare the SST gradients in the southwest Pacific sector over the
last glacial-interglacial cycle (Figure 6), we see an early cooling response between MIS 5e-d
corresponding to roughly half of the full glacial cooling, specifically in the cores located south of
the modern STF (for core list, see Table B1 in Appendix B). While not quantified, Bianchi &
Gersonde (2002) also describe a weakening of meridional SST gradients between the
Subantarctic and Antarctic Zones during MIS 5d in the Atlantic sector. Although this analysis is
based on sparse data, our SST reconstructions are consistent with the idea that surface ocean
cooling, a weakening of meridional SST gradients, and changes to the overall air-sea
disequilibrium could be responsible for at least some portion of the early $CO_2$ drawdown.



Further SST estimates from the region, and from the global ocean, are needed to substantiate
this hypothesis.

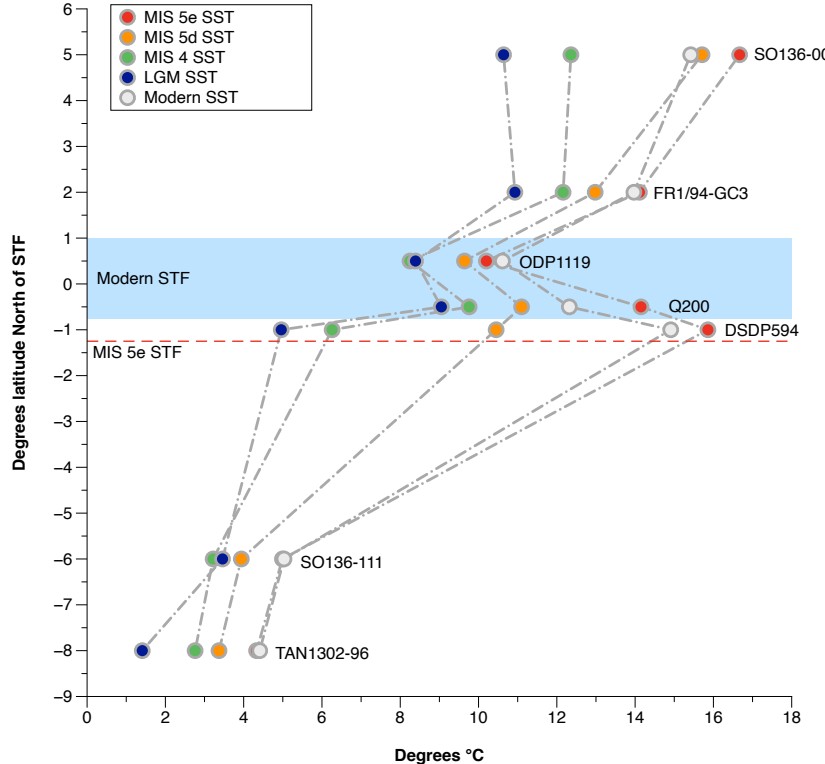

**Figure 6:** SST estimates from 7 cores located in the southwestern Pacific. SSTs used were 5-
point averages (depending on sampling resolution) taken at MIS peaks/median dates in
accordance with boundaries outlined in Lisiecki & Raymo, (2005). Due to the complex
circulation and frontal structures in the region, cores were plotted in +/- distance from the
average position of the modern STF. Cores used include: SO136-GC3 (SSTs calculated from
alkenones, Pelejero et al., 2006); FR1/94-GC3 (alkenones, Pelejero et al., 2006); ODP 181-1119
(PF-MAT, Hayward et al., 2008); DSDP594 (PF-MAT, Schaefer et al., 2005); Q200 (PF-MAT,
Weaver et al., 1998); SO136-111 (D-MAT, Crosta et al., 2004); and TAN1302-96 (D-MAT; *this
study*).


### 4.4 Sea Ice Expansion and Ocean Circulation

Although the TAN1302-96 wSIC record suggests that sea ice was largely absent at the

core site until the mid-glacial (~65 ka), the observed changes in sea ice throughout the glacial-

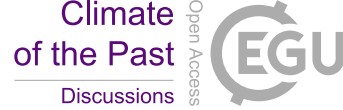

interglacial cycle may be linked to regional fluctuations in Antarctic Intermediate Water (AAIW)
subduction. The annual growth and decay of Antarctic sea ice plays a critical role in regional
water mass formation. Brine rejection results in net buoyancy loss in regions of sea ice
formation, while subsequent melt results in freshwater inputs and net buoyancy gains near the
ice margin (Shin et al., 2003; Pellichero et al., 2018). This increased freshwater input and
buoyancy gain near the ice margin can hinder AAIW subduction, with direct and indirect
impacts on both the upper and lower branches of the meridional overturning circulation
(Pellichero et al. 2018).

Previous research has used $\delta^{13}$C in benthic foraminifera to track changes in the depth of

the interface between AAIW and Upper Circumpolar Deep Water (UCDW) (Pahnke and Zahn,
2005; Ronge et al., 2015). Low $\delta^{13}$C values are linked to high nutrient concentrations found at
depths below ~1500 m in the UCDW, and higher $\delta^{13}$C values are associated with the shallower
AAIW waters (Figure 5). Marine sediment core MD97-2120 (45.535°S, 174.9403°E, core depth
1210 m) was retrieved from a water depth near the interface between the AAIW and UCDW
water masses. Over the last glacial-interglacial cycle, fluctuations in the benthic $\delta^{13}$C values
from MD97-2120 suggest that the core site was intermittently bathed in AAIW and UCDW, and
that the vertical extent of AAIW fluctuated throughout the last glacial-interglacial cycle. Ronge
et al. (2015) used the $\delta^{13}$C values from MD97-2120 and other core sites to quantify the
contributions of AAIW to the waters overlying MD97-2120 (%AAIW, Appendix D). These results
suggest that during warm periods, MD97- 2120 exhibits more positive $\delta^{13}$C values,
corresponding to higher %AAIW, while cooler periods exhibit more negative values,
corresponding to lower %AAIW (Figure 5). This means that during cooler periods, the AAIW-
UCDW interface shoaled, reducing the total volume of AAIW and indirectly causing an
expansion of UCDW (Ronge et al., 2015).

Our comparison between %AAIW and regional wSIC estimates suggest a strong link

between the two (Figure 5). Specifically, we observe that sea ice expansion occurs during time
periods when AAIW has shoaled and UCDW has expanded (i.e., %AAIW is low). In contrast,
during periods of low wSIC and warmer summer sea surface temperatures (e.g., MIS 5e),
%AAIW is high. This correlation supports the idea that increased concentrations of regional sea

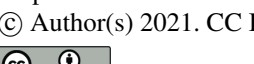



ice resulted in a substantial summer freshwater flux into the AAIW source region. This regional
freshening likely promoted a shallower subduction of AAIW and a corresponding volumetric
expansion of UCDW, which can be seen by the isotopic offset of the $\delta^{13}$C values between the
reference cores, and also by the increased carbonate dissolution in MD97-2120 during glacial
periods (Figure 7) (Pahnke et al., 2003; Ronge et al., 2015). These findings directly link sea ice
proxy records to observed changes in ocean circulation and water mass geometry.

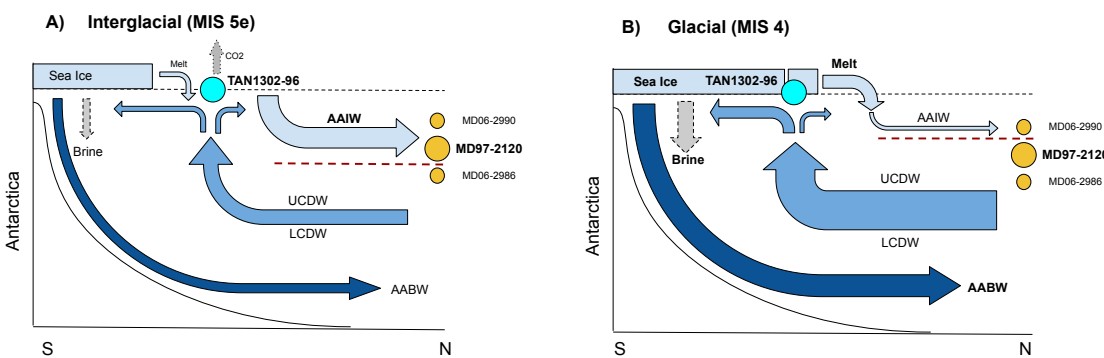

**Figure 7:** Schematic of changes in southwestern Pacific sector sea ice coverage and water mass
geometry between interglacial and glacial stages. **A)** Depicts interglacial conditions where sea
ice coverage is minimal and freshwater input from summer sea ice melt is low. This lack of
freshwater input allows AAIW to subduct to deeper depths and bath core MD97-2120,
capturing the higher $\delta^{13}$C signature of the overlying AAIW waters. The AAIW-UCDW interface
(red dashed line) is located beneath MD97-2120. $CO_2$ outgassing is occurring as carbon-rich
Circumpolar Deep Waters upwell near Antarctica. **B)** Depicts glacial conditions where sea ice
expansion has occurred beyond TAN1302-96, increasing brine rejection, and stabilizing the
water column. As a result of the increased sea ice growth, subsequent summer melt increases
the freshwater flux into the AAIW source region and increases AAIW buoyancy. This buoyancy
gain shoals the AAIW-UCDW interface above core MD97-2120, causing the core site to be
bathed in low $\delta^{13}$C UCDW. The shoaling of AAIW causes an indirect expansion of CDW,
increasing the glacial carbon stocks of the deep ocean while sea ice reduces $CO_2$ outgassing via
the capping mechanism.

In addition to its influence on regional freshwater forcing and AAIW reductions, these

sea ice changes may also coincide with larger-scale deep ocean circulation changes. The most
dramatic increases in winter sea ice observed in TAN1302-96 and SO136-111, along with
changes in %AAIW, are initiated during MIS 4. These shifts also correspond to basin-wide





changes in benthic $\delta^{13}C$ values in the Atlantic Ocean that suggest a shoaling in the Atlantic
Meridional Overturning Circulation (AMOC) during MIS 4 (Oliver et al., 2010; Kohfeld & Chase,
2017). Changes in deep ocean circulation are also recorded in $\varepsilon Nd$ isotope data in the Indian
sector of the Southern Ocean (Wilson et al., 2015), suggesting extensive changes in the AMOC
during this period. Recent modelling literature (Marzocchi & Jansen, 2019; Stein et al., 2020)
suggests that sea ice formation directly impacts marine carbon storage by increasing density
stratification and reducing diapycnal mixing, ultimately leading to $CO_2$ sequestration of an
estimated 30-40 ppm into the deep ocean. Taken collectively, the available data show that sea
ice expansion, AAIW-UCDW shoaling, changes in the AMOC, and a decrease in atmospheric $CO_2$
all occur concomitantly during MIS 4 (Figure 5). It appears likely, therefore, that sea ice
expansion during this time influenced intermediate water density gradients through increased
freshening and consequent shoaling of AAIW. This appears to have occurred while
simultaneously influencing deep-ocean density, and therefore stratification, through brine
rejection and enhanced deep water formation, which ultimately lead to decreased ventilation
(Abernathey et al., 2016). These changes in ocean stratification, combined with the sea ice
'capping' mechanism, appear to agree with both the recent modelling efforts (Stein et al., 2020)
and observed proxy data, and fit well within the hypothesis that mid-glacial $CO_2$ variability was
primarily the result of a more sluggish overturning circulation (Kohfeld & Chase, 2017).

**5.0 Summary & Conclusion**

This study presents new wSIC and sSST estimates from marine core TAN1302-96,

located in the southwestern Pacific sector of the Southern Ocean. We find that the wSIC
remained low during the early glacial cycle (130 to 70 ka), expanded during the middle glacial
cycle (~65 ka), and reached its maximum just prior to the LGM (~24.5 ka). These results largely
agree with nearby core SO136-111 but display some differences in wSIC magnitude with E27-
23. This discrepancy may be explained by differences in statistical applications and/or lateral
sediment redistribution, although more analysis is required to determine the exact cause(s).

The lack of changes in sSSTs and the absence of sea ice over the core site during the

early glacial suggests that the sea ice capping mechanism and corresponding surface



stratification in this region is an unlikely cause for early $CO_2$ drawdown, and that alternative
hypotheses should be considered when evaluating the mechanism(s) responsible for the initial
drawdown. More specifically, we consider the impact of changes in SST gradients between ~40°
to 60°S and support the idea that changes in air-sea disequilibrium associated with reduced
sea-surface temperature gradients could be a potential mechanism that contributed to early
glacial reductions in atmospheric $CO_2$ concentrations (Khatiwala et al., 2019). Another key
consideration is the potentially non-linear response between sea ice expansion and $CO_2$
sequestration potential (i.e., that not all sea ice is equal in its capacity to sequester carbon).
More analyses are required to adequately address this.

We also observe a strong link between regional sea ice concentrations and vertical

fluctuations in the AAIW-UCDW interface. Regional sea ice expansion appears to coincide with
the shoaling of AAIW, likely due to the freshwater flux from summer sea ice melt increasing
buoyancy in the AAIW formation region. Furthermore, major sea ice expansion and AAIW
shoaling occurs during the middle of the glacial cycle and is coincident with previously
recognized shoaling in AMOC and mid-glacial atmospheric $CO_2$ reductions, suggesting a
mechanistic link between sea ice and ocean circulation.

This paper has focused exclusively on sea ice as a driver of physical changes, but we

recognize that these changes in sea ice will be accompanied by multiple processes that interact
and compete with each other. Marzocchi & Jansen (2019) note that teasing apart the individual
components of $CO_2$ fluctuations is complicated because of interactions between sea ice
capping, air-sea disequilibrium, AABW formation rates, and the biological pump. We recognize
that these processes may not act independently, and as such, hope to contribute new data to
help advance our collective understanding of the role of sea ice on influencing atmospheric $CO_2$
variability on a glacial-interglacial time scale.

**6.0 Appendices**
**Appendix A: Age Model & Sampling Depths**
**Table A1:** Radiocarbon dates taken from TAN130-296. NDFB = Not Distinguishable from Background



| Lab Code | Sample Material | Core Name | Depth (cm) | δ13C (per mil) | δ13C (+/-) | % Modern Carbon | 1σ error | Fraction Modern | (+/-) | Radiocarbon Year | 1σ error | Reference |
|---|---|---|---|---|---|---|---|---|---|---|---|---|
| NZA 57105 | *N. pachyderma* and *G. bulloides* | TAN1302-96 | 21 | 1 | 0.2 | / | / | 0.5982 | 0.0018 | 4127 | 24 | Prebble et al., 2017 |
| NZA 57109 | *N. pachyderma* and *G. bulloides* | TAN1302-96 | 50 | 0.7 | 0.2 | / | / | 0.3723 | 0.0015 | 7936 | 32 | Prebble et al., 2017 |
| OZX 517 | *N. pachyderma* and *G. bulloides* | TAN1302-96 | 63 | 1 | 0.1 | 30.62 | 0.15 | / | / | 9505 | 40 | *This study* |
| NZA 61429 | *N. pachyderma* and *G. bulloides* | TAN1302-96 | 75 | 0.7 | 0.2 | / | / | 0.2373 | 0.0011 | 11554 | 37 | Prebble et al., 2017 |
| OZX 518 | *N. pachyderma* and *G. bulloides* | TAN1302-96 | 87 | -0.1 | 0.1 | 19.62 | 0.11 | / | / | 13085 | 45 | *This study* |
| OZX 519 | *N. pachyderma* and *G. bulloides* | TAN1302-96 | 130 | 1.7 | 0.1 | 0.02 | 0.04 | / | / | NDFB | / | *This study* |
| OZX 520 | *N. pachyderma* and *G. bulloides* | TAN1302-96 | 170 | -1.1 | 0.3 | 0.03 | 0.04 | / | / | NDFB | / | *This study* |

**Table A2:** Tie points used in construction of the TAN1302-96 age model

| TAN1302-96 Depth (cm) | TAN1302-96 δ18O | LR04 Age | LR04 δ18O |
|---|---|---|---|
| 110 | 4.710 | 18000 | 5.02 |
| 170 | 3.930 | 56000 | 4.35 |
| 200 | 3.782 | 70000 | 4.32 |
| 260 | 3.150 | 115000 | 3.71 |
| 300 | 3.660 | 129000 | 3.9 |
| 320 | 4.350 | 140000 | 4.98 |

**Table A3:** Sample depth and corresponding age. Diatom slides using Method 1 used sediment samples that are
even (e.g., 10, 20, 30, etc.), while diatom slides using Method 2 used sediment samples that are odd (e.g., 53, 87,
etc.). * Indicates the sample was calculated based on linear sedimentation rates.

| Sample Depth (cm) | Age | Sample Depth (cm) | Age | Sample Depth (cm) | Age | Sample Depth (cm) | Age |
|---|---|---|---|---|---|---|---|
| 10 | 1802* | 100 | 16167 | 197 | 68849 | 260 | 114169 |
| 20 | 3282* | 103 | 16720 | 200 | 70541 | 263 | 115690 |
| 30 | 4762 | 107 | 17784 | 203 | 72417 | 267 | 117398 |
| 40 | 6252 | 110 | 18818 | 207 | 75211 | 270 | 118536 |
| 50 | 7736 | 113 | 20200 | 210 | 77358 | 273 | 119565 |
| 53 | 8168 | 117 | 22364 | 213 | 79629 | 277 | 120909 |
| 57 | 8727 | 120 | 24202 | 217 | 82635 | 280 | 121881 |
| 60 | 9147 | 123 | 26113 | 220 | 84965 | 283 | 122875 |
| 63 | 9566 | 127 | 28672 | 223 | 87235 | 287 | 124228 |
| 67 | 10469 | 130 | 30565 | 227 | 90337 | 290 | 125296 |
| 70 | 11175 | 140 | 37018 | 230 | 92635 | 293 | 126338 |
| 73 | 11910 | 150 | 43401 | 233 | 94922 | 297 | 127857 |
| 77 | 12716 | 160 | 49744 | 237 | 97910 | 300 | 129151 |



| 80 | 13103 | 170 | 55709 | 240 | 100180 | 303 | 130544 |
|----|-------|-----|-------|-----|--------|-----|--------|
| 83 | 13504 | 180 | 60640 | 243 | 102430 | 307 | 132573 |
| 87 | 14065 | 183 | 62021 | 247 | 105398 | 310 | 134377 |
| 90 | 14584 | 187 | 63905 | 250 | 107642 | 313 | 136156 |
| 93 | 15066 | 190 | 65302 | 253 | 109797 | 317 | 138193 |
| 97 | 15698 | 193 | 66737 | 257 | 112475 | 320 | 139591 |


## Appendix B: Supporting Information

**Table B1:** Information for all cores used in calculating southwestern Pacific sector SST gradients (Figure 7).

| Core Name | Latitude | Longitude | Depth | Age Model Reference | Data Used | Data Source |
|-----------|----------|-----------|-------|---------------------|-----------|-------------|
| **TAN1302-96** | 59.09°S | 157.05°E | 3099 m | *This study* | n/a | *This study* |
| **SO136-111** | 56.66°S | 160.23°E | 3912 m | Crosta et al., 2004 | wSIC; SST | Crosta et al., 2004; *This study* |
| **SO136-GC3** | 42.3°S | 169.88°E | 958 m | Pelejero et al., 2006; Barrows et al., 2007 | δ13C; SST | Pelejero et al., 2006; Ronge et al., 2015 |
| **FR1/94-GC3** | 44.25°S | 149.98°E | 2667 m | Pelejero et al., 2006 | SST | Pelejero et al., 2006 |
| **ODP 1119-181** | 44.75°S | 172.39°E | 396 m | Wilson et al., 2005 | SST | Wilson et al., 2005; Hayward et al., 2008 |
| **DSDP 594** | 45.54°S | 174.94°E | 1204 m | Nelson et al., 1985; Kowalski & Meyers 1997 | SST | Schaefer et al., 2005 |
| **Q200** | 45.99°S | 172.02°E | 1370 m | Waver et al., 1998 | SST | Weaver et al., 1998 |

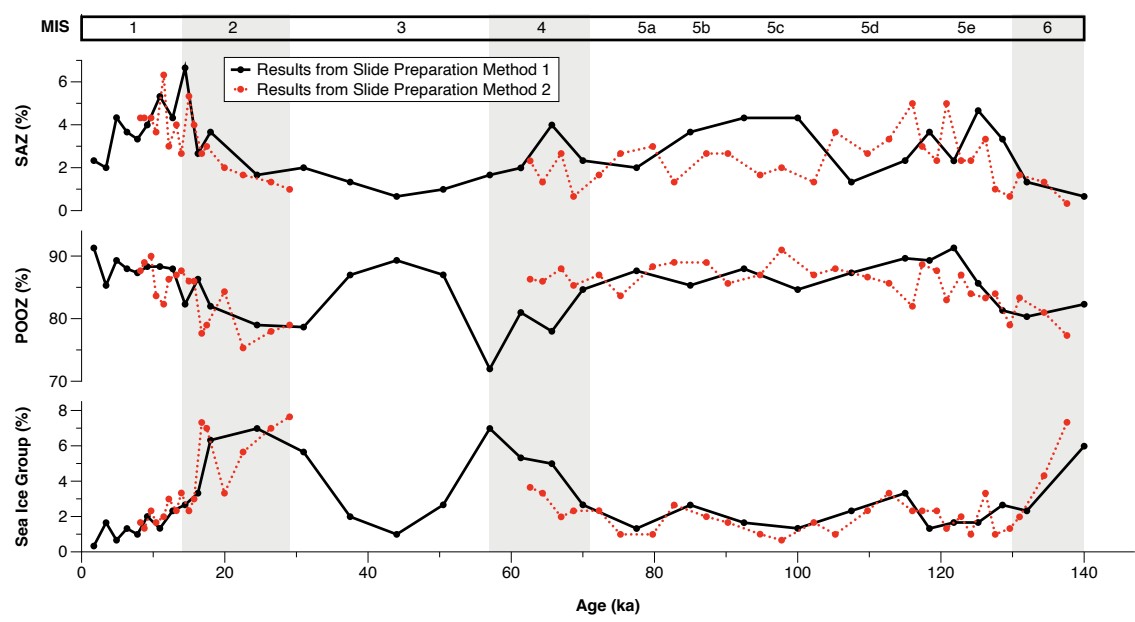



**Figure B1:** Results from diatom slide preparation methods 1 & 2. No notable differences or biases were observed
between the two different methods.

## Appendix C: TAN1302-96 and E27-23 Comparison
**Potential Causes for wSIC Estimate Differences**

The first potential cause for the observed differences between TAN1302-96 and E27-23

wSIC estimates is through the cumulative effects of different laboratory protocols. While it is
difficult to determine precisely how much different laboratory protocols could influence the
results, we cannot exclude this explanation as a possible contributor to differences in wSIC.

The second potential cause for differences in wSIC estimates between E27-23 and

TAN1302-96 are differences in counting and identification methods. We believe this is an
unlikely cause for the differences observed between E27-23 and TAN1302-96 primarily because
of the magnitude of counting discrepancies required to cause a difference of 50% wSIC
estimates between the two cores. The close coupling of wSIC estimates between TAN1302-96
and SO136-111 over the entire glacial-interglacial cycle supports that a fundamental issue
relating to taxonomic identification and/or methodology is an unlikely explanation for the
observed wSIC differences.

Finally, the fourth potential cause of differing wSIC estimates is selective diatom

preservation (e.g., Pichon et al., 1999; Ragueneau et al., 2000). The similarities between
TAN1302-96 and SO136-111 wSIC estimates, along with independent indicators in cores E27-23
and TAN1302-96, suggest that this is unlikely. For E27-23, Bradtmiller et al. (2009) used the
consistent relationship between $^{231}$Pa/$^{230}$Th ratios and opal fluxes to suggest that dissolution
remained relatively constant between the LGM and Holocene periods. In TAN1302-96, we
assigned a semi-quantitative diatom preservation value between 1 (extreme dissolution) and 4
(virtually perfect preservation) for each counted specimen. The average preservation of
diatoms for the entire core was 3.38 ± 0.13, with no observed bias based on sedimentation rate
or MIS. This assessment, although semi-qualitative, suggests that preservation remained
relatively constant (and good) throughout TAN1302-96, and is therefore unlikely to cause large
differences in wSIC between the two cores.



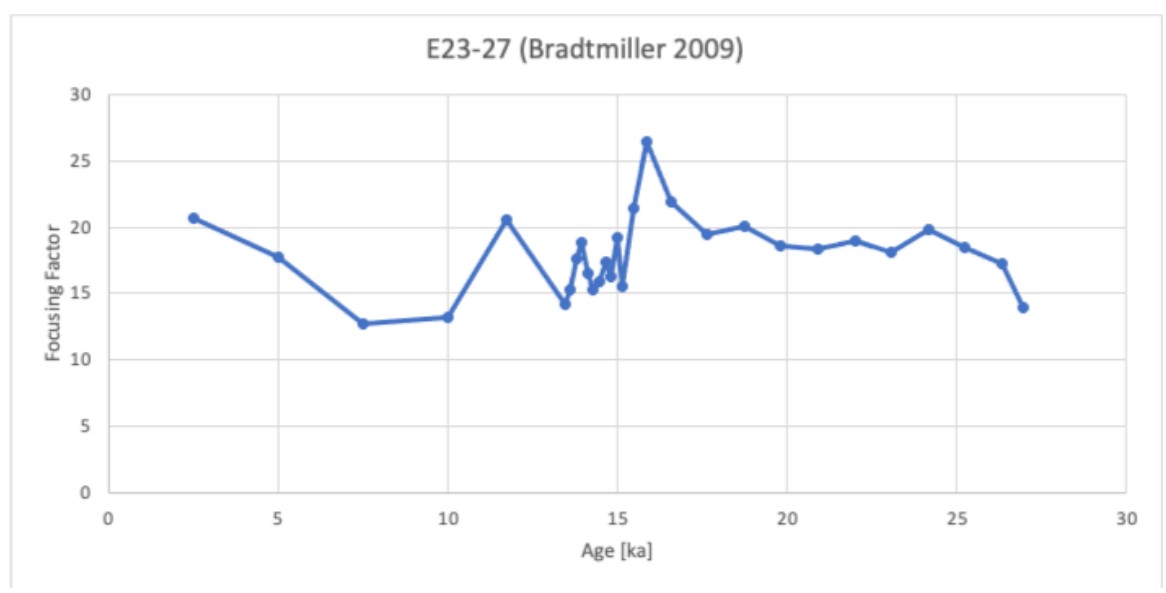

**Figure C1:** Preliminary focusing factor (FF) values for E27-23. These results suggest notable lateral sediment redistribution over the last 26 ka, requiring further analysis (Bradtmiller et al., 2009).

## Appendix D: %AAIW Calculation

The calculation of %AAIW in this study is the same as was used in Ronge et al. (2015):

$$\%AAIW = (\delta^{13}C_{MD97\text{-}2120} - \delta^{13}C_{MD06\text{-}2986}) / (\delta^{13}C_{MD06\text{-}2990} - \delta^{13}C_{MD06\text{-}2986}) * 100$$

All core information for MD97-2120, MD06-2986, and MD06-2990, along with supporting supplemental information can be found through the original publication.

## 7.0 Data Availability

All data has been submitted to Pangaea (PDI-29255) and is awaiting publication. Once Pangaea has published the dataset, the corresponding author will supply the DOI.

## 8.0 Author Contributions

The authors confirm that the contributions to this paper are as follows: study conception and design: KK, HB; author data collection: JJ, KK, HB, XC, ML, GD, ZC, AL; analysis and interpretation of results: JJ, KK, HB, XC, ZC, AL, HA, GJ; draft manuscript preparation and/or editing: JJ, KK, HB,



XC, GD, ZC, AL, HA, GJ. All authors reviewed the results and approved the final version of the
manuscript.

**9.0 Competing Interests**
The authors declare that they have no conflict of interest.

**10.0 Acknowledgements**

This work was supported by a Canadian Natural Sciences and Engineering Research

Council grant (Discovery Grant RGPIN342251) to Karen Kohfeld. Travel funding for workshop
collaboration was provided to Jacob Jones by a Past Global Changes (PAGES) grant to the Cycles
of Sea Ice Dynamics in the Earth System (C-SIDE) Working Group. Rachel Meyne (Colgate
University) assisted with slide preparation; Maureen Soon (University of British Columbia)
assisted with opal concentration measurements; Marlow Pellatt (Parks Canada) assisted with
project conceptualization and guidance. The TAN1302-96 core was collected during the
TAN1302 RV Tangaroa voyage to the Mertz Polynya. We would like to thank the Voyage leader
Dr. Mike Williams and Captain Evan Solly and the crew, technicians and scientists involved in
the TAN1302 voyage. The voyage was co-funded by NIWA, Australian, and French research
funding. We acknowledge Dr. Andrew Kingston for running the stable isotopes at NIWA. We
acknowledge ANSTO grant AP11676 for funding the additional radiocarbon dates. This research
was partially supported by the Australian Government through the Australian Research
Council's Discovery Projects funding scheme (project DP180102357, awarded to Zanna Chase
and Helen Bostock).

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
