# Peer review of "Sea Ice Changes in the Southwest Pacific Sector of the Southern Ocean During the Last 140,000 Years"

_Climate of the Past, 2021_

## Referee Comment (RC1)

General comments:

The paper addresses very important issue of the long-term climate changes and the interaction between sea ice growth, ocean circulation and CO2 concentrations in the atmosphere.

The dataset uses diatom-based Modern Analogue Technique to reconstruct SST and SIC utilizing a large dataset of surface sediment samples from the Southern Ocean. The problem of past oceanic conditions is handled in a broad perspective, analyzing other core data in the region. Overall, the paper is very interesting and very clearly written.

However, there are several issues corresponding to the sediment cores described in this study and the presentation of their results. These are listed below:

(1) Methods, section 2.1 – it's not entirely clear which cores are recalculated for SIC and which for SST as part of this study; I think all the cores should be mentioned in the methods and clarified which are analysed from the scratch and which have had their results recalculated; and which are jut cited. Caption to Figure 1 is confusing in this matter.

(2) Results – what is missing here is the figure and description for the results of the recalculated core SO136-111; it is a part of this study and needs to be described.

(3) Discussion, section 4.1 – this part belongs to Results, not in Discussion and the sentences that do belong to Discussion should just briefly describe the past conditions and trends for SIC and SST, e.g. line 276-278 and 282-283 so I suggest restructuring. And please provide time intervals for the periods you describe in text. Also, there is first mention of the core E27-23, which was not mentioned in Methods or Results and if it is a part of recalculation then it should be properly described. Otherwise, please provide a citation for this core.

The paper can be accepted after revision, which is mainly centered on proper presentation of all analysed/recalculated core data in Methods and Results, as well clarity regarding the types of analyses done on these cores.

Specific comments:

Abstract

l. 26 – what quantitative technique was used to reconstruct SIC and SST? Transfer function? Please clarify

l. 30 – please provide percentage info for the SIC (consolidated)

Abstract overall – seems like there might be too much detail regarding the past conditions, could be simplified and generalized a bit, e.g. SST values could be mentioned only for the minimum and maximum values and otherwise just refer to trends

Introduction

l. 51 – what does it mean 'dynamically linked'?

l. 66 – either 23 to 19 ka or 23.000 to 19.000, please use consistent time scale; also is it BP?

l. 64-84 – are there any other proxies providing information on reconstructed oceanic variability in the region? Such as foraminifera etc? Would be nice to mention

l. 88-92 – this belongs to methods; introduction should mainly state general information on the materials studied

Methods

2.2. – very nice with the 2-method approach!

l. 196-206 – this part belongs to Results section

l. 201 – what is the sea ice concentration range for this group?

Table 1 – just curious, did you identify any Thalassiosira antarctica var. antarctica? Its northern equivalent is pretty common in the Arctic and sub-Arctic region

l. 212 – why did you choose this period only? Is the present-day diatom succession limited to January-March? Please clarify

l. 227-228 – it would be nice to consider other quantitative, transfer function tests at some point, such as ML (MLRC) and WA-PLS to show that MAT is indeed the best choice.

Results

l. 242 – which periods specifically? Looks like MIS 1, 4 and 5

Discussion

l. 289 – I can't find the description of cores MD06 in Discussion

Line 308 – please clarify that you mean explanation no. 3

Line 308-330 – I suggest to put this text in a separate sub-chapter as it stands out of the description of past conditions

Section 4.2. – I like the description!

l. 462 – reference for the core is needed here

l. 528 – again, if these two cores are a key element of Discussion and overall conclusions, then we need more info in Methods and Results

l. 549-556 – perhaps this part fits better to Introduction

Technical corrections:

l. 29 - coordinates etc should be removed from Abstract, too much detail

l. 36 – '…coolest values, respectively…'

l. 38 – SSST – too many S or sSSTs

l. 87 – SSSTs – should it be singular?

l. 114 – 'published cores providing recalculated sea ice extent data'?

Figure 1 – please add abbreviations SSI and WSI in legend

---

## Author Comment (AC2)

WSI % - Ferry et al. 2015 (Blue) vs MAT249 (Orange)

---

## Author Comment (AC3)

TAN1302-96 SSST (blue) vs SD (orange)

---

## Author Response (AR1)

**Referee Comment Tracking Table for:**

**Sea Ice Changes in the Southwest Pacific Sector of the Southern Ocean During the Last 140,000 Years**

Jacob Jones[1], Karen E Kohfeld[1,2], Helen Bostock[3,4], Xavier Crosta[5], Melanie Liston[6], Gavin Dunbar[6], Zanna Chase[7], Amy Leventer[8] Harris Anderson[7], Geraldine Jacobsen[9]

[1] School of Resource and Environmental Management, Simon Fraser University, Burnaby, Canada
[2] School of Environmental Science, Simon Fraser University, Burnaby, Canada
[3] School of Earth and Environmental Sciences, The University of Queensland, Brisbane, Australia
[4] National Institute of Water and Atmospheric Research (NIWA), Wellington, New Zealand
[5] Université de Bordeaux, CNRS, EPHE, UMR 5805 EPOC, Pessac, France
[6] Antarctic Research Centre, Victoria University of Wellington, Wellington, New Zealand
[7] Institute of Marine and Antarctic Studies, University of Tasmania, Hobart, Australia
[8] Geology Department, Colgate University, Hamilton, NY, USA
[9] Centre for Accelerator Science, Australian Nuclear Science and Technology Organisation, Lucas Heights, NSW, Australia

*Correspondence to*: Jacob Jones (jacob_jones@sfu.ca)

**Author's Response:**
We would like to thank the reviewers and editor for their time considering this manuscript. We have reviewed all comments received and provided updates to our initial responses. We appreciate the comments provided to us, as they have allowed us to refine the manuscript and provide a more robust age model for marine core TAN1302-96. Using the table below, we indicate where in the manuscript these comments have been addressed, and provided the specific language used (where applicable). For those responses that provided clarity to the reviewers and did not require changes to the manuscript, the responses are the same for the initial and final responses.

Please note that the line numbers have changed from past comments due to the addition (or removal) of some text. New line numbers have been provided for each comment in the 'Author's Final Response + Line Number' column.

| Comment | Comment | Author's Initial Response | Author's Final Response + Line Number |
|---|---|---|---|
| RC1-1 | Methods, section 2.1 – it's not entirely clear which cores are recalculated for SIC and which for SST as part of this study; I think all the cores should be mentioned in the methods and clarified which are analysed from | Agreed – we will add all cores to methods and provide more clarity around which cores provide what information, and where this information was collected from. | New line number(s): 336-363; 173-180

We have added an additional section (2.4 – Additional Core Data). This section also includes Table 2 (line 363), which includes all cores used |

| | | For reference: | throughout this analysis and all relevant information. |
|---|---|---|---|
| | the scratch and which have had their results recalculated; and which are jut cited. Caption to Figure 1 is confusing in this matter | TAN96 => new data both for SSST and WSIC | |
| | | SO136 => WSIC recalculated through augmented modern database (249 analogs vs 195 analogs in Crosta 2004) | We have also included a reference to Table 2 within the Figure 1 (lines 173-180) caption to reduce confusion. |
| | | E27-23 => Published data (Ferry et al., 2015) | |
| RC1-2 | Results – what is missing here is the figure and description for the results of the recalculated core SO136-111; it is a part of this study and needs to be described. | Agreed – we will include a description of the recalculated SO136-111 results. | New line number(s): 433-449

We have added an additional results section (3.3 – SO136-111 SSST and WSIC Recalculation). This section describes the results from the recalculation of SO136-111. |
| RC1-3 | Discussion, section 4.1 – this part belongs to Results, not in Discussion and the sentences that do belong to Discussion should just briefly describe the past conditions and trends for SIC and SST, e.g., line 276-278 and 282-283 so I suggest restructuring. And please provide time intervals for the periods you describe in text. Also, there is first mention of the core | Agreed – we will restructure this section to fit into Results and will provide time intervals described in text.

E27-23 is not recalculated within this study, but a citation (Ferry et al., 2015) will be provided. | New line number(s): 471-480

We have restructured and removed unnecessary sentences that do not belong in the Discussion. The Discussion has been streamlined such that only a brief description of the past conditions and trends is present. |

| | | | |
|---|---|---|---|
| | E27-23, which was not mentioned in Methods or Results and if it is a part of recalculation then it should be properly described. Otherwise, please provide a citation for this core. | | |
| RC1-4 | l. 26 – what quantitative technique was used to reconstruct SIC and SST? Transfer function? Please clarify | We used a diatom-based transfer function. We will update the language and clarify accordingly. | New line number(s): 26-29

We have included more information regarding the quantitative techniques used. The text now reads:

"Here we provide new estimates of winter sea ice concentrations (WSIC) and summer sea surface temperatures (SSST) for a full glacial-interglacial cycle from the southwestern Pacific sector of the Southern Ocean using the Modern Analog Technique (MAT) on fossil diatom assemblages…" |
| RC1-5 | l. 30 – please provide percentage info for the SIC (consolidated) | Generally, 0-15% open ocean; 15-40% unconsolidated sea ice; >40% consolidated sea ice (Armand et al., 2005 and references therein).

We will clarify the text to something like:

"Following the modern concept (Armand et al., 2005 and references | New line number(s): 19-41; 330-332

We have reworked the Abstract such that no mention of consolidated (or unconsolidated) sea ice is present and clarity is no longer required here.

We have provided references for the |

| | | | |
|---|---|---|---|
| | | therein), we find that winter sea ice was consolidated (wSIC = >40%) over the core site …" | percentages for SIC later in the manuscript (lines 330-332) as follows:

"As outlined in Ferry et al., (2015), we consider <15% WSIC to represent an absence of winter sea ice, 15-40% WSIC as present but unconsolidated, and >40% to represent consolidated winter sea ice." |
| RC1-6 | Abstract overall – seems like there might be too much detail regarding the past conditions, could be simplified and generalized a bit, e.g. SST values could be mentioned only for the minimum and maximum values and otherwise just refer to trends | Agreed – we will rework to remove excess details (e.g., lat/long coordinates and water depth) and align with other comments regarding the Abstract (e.g., RC1-23, 24). | New line number(s): 19-41

We have reworked and streamlined the Abstract to remove unnecessary detail (incl. lat/long, water depth, etc.). |
| RC1-7 | l. 51 – what does it mean 'dynamically linked'? | We use the term 'dynamic' in the convention sense to describe a "force that controls or influences a process of growth, change, interaction or activity" (from Merriam-Webster).

We therefore describe the link between sea ice and carbon sequestration as being 'dynamically' linked because each factor exerts some force or influence over the other. | |
| RC1-8 | l. 66 – either 23 to 19 ka or 23.000 to 19.000, please use consistent time scale; also, is it BP? | Yes – ages are presented in BP.

We will update and standardize to 'ka BP' throughout. | New line number(s): throughout

We have updated and standardized all dates to be 'ka BP' throughout the manuscript. |

| | | | |
|---|---|---|---|
| RC1-9 | l. 64-84 – are there any other proxies providing information on reconstructed oceanic variability in the region? Such as foraminifera etc? Would be nice to mention | Yes - there are other proxy reconstructions from the region that provide information on oceanic variability (dust, nitrogen, temperature, etc.), but not many foraminifera reconstructions. However, these proxies don't necessary look at sea ice variability or capture it in the same capacity as do the use of diatoms and transfer functions.

We've kept this paragraph and paper primarily focused on sea ice, and while we acknowledge and appreciate the reviewer's suggestion to include other key proxy reconstructions from the region, we feel as though discussing additional proxies from the region may detract from the tightly focused narrative.

It is also worth mentioning that forthcoming submissions from Chadwick et al. and Kohfeld et al., which will be submitted to this special issue, will address some of the larger topics concerning regional oceanic variability and reconstructions from the region and will supplement this manuscript. | |
| RC1-10 | l. 88-92 – this belongs to methods; introduction should mainly state general information on the materials studied | After some consideration, the co-authors have agreed that the text provided on lines 88-92 would likely be useful to readers who skip the methods section and only quickly read the paper.

We will remove unnecessary information (e.g., latitude & longitude, water depth) to streamline the reading, but believe the additional references to SO136-111 and E27-23 should remain. | New line number(s): 131-134

We have decided to keep the reference to SO136-111 and E27-23, including the lat/long and water depth, as this information may prove useful for readers who do not read the paper in its entirety. We believe that the information included does not detract from reading the manuscript; however, we are happy to work with the referees/editor to cut back as they see fit. |
| RC1-11 | l. 196-206 – this part belongs to Results section | The text provided in the manuscript on line 196 may be slightly | New line number(s): 288-289 |

| | | misleading – our analysis did not establish these taxonomic groups, as these have been used in other publications and are established methods (e.g., Crosta et al., 2004, Ghadi et al., 2020). We will therefore change the wording to something like: "Based on previously established taxonomic groups, diatoms were grouped into one of three categories based on temperature preference and sea ice tolerance: …" | We have updated the text to read: "Based on previously established taxonomic groups (Crosta et al., 2004), diatoms were grouped into one of three categories based on temperature preference and sea ice tolerance…". |
|---|---|---|---|
| RC1-12 | l. 201 – what is the sea ice concentration range for this group? | The highest abundances of the diatom species composing this group in the modern sediments are found at WSI greater than 60-70% (Zielinski and Gersonde, 1997; Armand et al., 2005; Esper et al., 2010). They are therefore all suited to record past changes in WSI (Esper et al., 2014). | |
| RC1-13 | Table 1 – just curious, did you identify any Thalassiosira antarctica var. antarctica? Its northern equivalent is pretty common in the Arctic and sub-Arctic region | Only a few specimens of Thalassiosira antarctica var antarctica form 2 (warm variety; Taylor et al., 2002) were found, and they were generally identified during glacial periods. TAN => up to 1% of the total diatom assemblages SO136 => up to ~2% of the total diatom assemblages E27-23 => up to 1.5% of the total diatom assemblages | |
| RC1-14 | l. 212 – why did you choose this period only? Is the present-day diatom succession limited to January-March? Please clarify | January-March is mentioned only for the SST. In the Southern Ocean, diatom production is restricted to the sunlit period (spring to fall). | New line number(s): 312-314 We have updated the text to read: |

| | | Production starts earlier in the SAZ-POOZ than in the Sea Ice Zone, which is especially late in the coastal zone due to high sea-ice cover (Nelson et al., 2001; Arrigo et al., 2004; Grigorov et al., 2014). | "Summer (January to March) SST was estimated because it is considered to be a better explanatory variable than spring or annual SST (Esper et al., 2010; Esper & Gersonde, 2014b)." |
|---|---|---|---|
| | | Although there is a succession in diatom production from spring to fall (Grigorov et al., 2014) and that spring production may exceed summer production in some regions (Fiala et al., 2002), most of the export occurs during the summer months (Fiala et al., 1998; Kopczynska et al., 1998; Fischer et al., 2002; Armand et al., 2008; Grigorov et al., 2014; Rigual-Hernandez et al., 2015). | |
| | | For these reasons, summer SST is generally a better explanatory variable than spring or annual one (Esper et al., 2014). | |
| | | We will add additional clarity (not to this degree) to the manuscript to resolve any confusion. | |
| RC1-15 | l. 227-228 – it would be nice to consider other quantitative, transfer function tests at some | Other transfer functions have been tested using the modern diatom database used here (Ferry et al., | |

| | point, such as ML (MLRC) and WA-PLS to show that MAT is indeed the best choice. | 2015) and using another modern diatom database (Esper et al., 2014). | |
|---|---|---|---|
| | | In Esper et al. (2014), MAT performed better in term of $R^2$ and RMSEP. Though the G-IG patterns were reconstructed with both IKM and MAT, the latter reconstructed more variable sea ice at the multi-millennial timescale as IKM is known to smooth down records due to its approach (regression and paleo-environmental equation; Esper et al., 2014). Conversely, GAM and MAT provided similar results in core SO136-111 (Ferry et al., 2015). | |
| | | Finally, it is worth noting that MAT provides SST and WSI reconstructions that are in agreement with other type of SST and WSI reconstructions (Gersonde et al., 2005; Civel et al., 2021), other downcore proxies and, more globally, Southern Ocean paleoclimate at any timescales (Crosta et al., 2004; Nair et al., 2019; Ghadi et al., 2020; Orme et al., 2020; Crosta et al., 2021; Shukla et al., 2021). | |
| | | This topic will also be discussed in the forthcoming Kohfeld et al. manuscript, which will be submitted to the same special issue. | |
| RC1-16 | l. 242 – which periods specifically? Looks like MIS 1, 4 and 5 | We will update wording to something like:

"The Sub-Antarctic Zone (SAZ) group had relatively low abundances, with higher values occurring generally during the warmer interstadial periods MIS 1 and 5, and briefly during MIS 4 at 67 ka."' | New line number(s): 374-375

We have updated the text to read:
"The Sub-Antarctic Zone group had relatively low abundances, with higher values occurring during warmer interstadial periods (MIS 5 and the Holocene) and briefly during MIS 4 at ~65 ka BP." |
| RC1-17 | l. 289 – I can't find the description of cores MD06 in Discussion | We will rework the Methods & Results section to include a description of all cores | New line number(s): 335-363 |

| | | that were used in the manuscript that were not already introduced (in line with comment RC1-21). | We have added Section 2.4 (Additional Core Data), which includes Table 2 (line 363). This table includes reference to all cores used, including the MD06 cores. |
|---|---|---|---|
| | | The cores that will be introduced for the %AAIW calculation include: | |
| | | [1] MD06-2990; | |
| | | [2] MD06-2989; and | |
| | | [3] MD97-2120 | |
| | | From Pahanke & Zahn (2005) & Ronge et al. (2015). | |
| | | In addition, the following cores are used in Discussion 4.3 for the SST gradient: | |
| | | [1] SO136-GC3; | |
| | | [2] FR1/94-GC3; | |
| | | [3] ODP1119; | |
| | | [4] Q200; and | |
| | | [5] DSDP594. | |
| | | We will add a sentence in the manuscript that points to these cores (and references) so that all cores used in this analysis are included within the text and cited. | |
| RC1-18 | l. 308 – please clarify that you mean explanation no. 3 | Noted - we will rearrange the numbering as follows and update | New line number(s): 515-572 |

| | | lines 305-307 accordingly: [1] Different statistical applications; [2] lateral sediment redistribution; [3] differences in laboratory protocols; [4] differences in diatom identification/counting methodology; and [5] selective diatom dissolution; We will then correct the numbering in the appendices but leave the wording from lines 308-327 as is. | We have changed the numbering of possible explanations as follows: [1] Different statistical applications; [2] lateral sediment redistribution; [3] differences in laboratory protocols; [4] differences in diatom identification/counting methodology; and [5] selective diatom dissolution; We believe these numbers provide clarity on our specific arguments. These numbers have also been updated in the Appendix. |
|---|---|---|---|
| RC1-19 | l. 308-330 – I suggest to put this text in a separate sub-chapter as it stands out of the description of past conditions | We initially had this section broken out as its own sub-chapter (as suggested), but after reading and having discussions around the chapter's flow, we decided to embed part of the discussion within the text and append the non-essential part of the discussion. We are open to reworking this section and separate it into a sub-chapter if the reviewer feels this is important; however, in our own writing/re-writing exercises we have found the current state of the manuscript to have the best reading flow. | |
| RC1-20 | l. 462 – reference for the core is needed here | Noted - we will add the Pahnke & Zahn (2005) reference for core MD97-2120. | New line number(s): 768 We have provided the Pahnke & Zahn (2005) reference for MD97-2120. |
| RC1-21 | l. 528 – again, if these two cores are a key | Noted - we will include a description of these | New line number(s): 363 |

| | | cores in previous sections. See response to RC1-17 for more information. | We have added Table 2 into the Methods section of the manuscript which includes references to all cores used. |
|---|---|---|---|
| RC1-22 | l. 549-556 – perhaps this part fits better to Introduction | We agree that the current reading of this paragraph would fit better within the Introduction. We would like to keep these ideas at the end of the paper, so we will rework the paragraph to read more as a conclusion.

We will change the text to read something like:

"In conclusion, this paper has focused exclusively on sea ice as a driver of physical change…"

and

"We recognize that these processes may not act independently, and as such, have contributed new data to help advance our collective understanding…" | New line number(s): 881; 886-891

We have updated the text to read:

"In conclusion, this paper has focused exclusively on sea ice as a driver…"

And

"We recognize that these processes may not act independently, and as such, have contributed new data to help advance our collective understanding…" |
| RC1-23 | l. 29 - coordinates etc should be removed from Abstract, too much detail | Agreed - coordinates and water depth have been removed from the Abstract. | New line number(s): 19-41

We have reworked the Abstract to be in line with this comment and others. Coordinates and additional details have |

| | | | been removed to streamline reading. |
|---|---|---|---|
| RC1-24 | l. 36 – '…coolest values, respectively…' | Agreed – Line 36 now reads:

"WSIC and SSSTs reached their maximum concentrations and coolest values, respectively, by 24.5 ka…" | New line number(s): n/a

We have reworked the Abstract to be in line with other comments and as a result, this sentence is no longer included. |
| RC1-25 | l. 38 – SSST – too many S or sSSTs | This was a typing error and should have been "sSST"; however, based on other comments, we are updating all "sSST" to "SSST" throughout the manuscript. | New line number(s): n/a

This sentence has been removed from the Abstract and no longer requires this change.
We have also carefully proofread the manuscript for other similar typing errors. |
| RC1-26 | l. 87 – SSSTs – should it be singular? | Yes – SSST should be singular. The text currently reads:

"SSSTs and wSIC are estimated by applying the Modern Analogue Technique …"

We will change the wording to:

"SSST and WSIC estimates are produced by applying the Modern Analog Technique …" | New line number(s): 127-128

We have corrected the text to read:

"WSIC, which is a grid-scale observation of the mean state fraction of ocean area that is covered by sea ice over the sample period, and SSST estimates are produced by applying the Modern Analog Technique (MAT) to fossil diatom assemblages from sediment core TAN1302- |

| | | | 96 (59.09°S, 157.05°E, water depth 3099 m)" |
|---|---|---|---|
| | | | We have also carefully proofread the manuscript for other similar errors. |
| RC1-27 | l. 114 – 'published cores providing recalculated sea ice extent data'? | Only SO136-111 has been recalculated for this study. We will update the language to: "… and additional published cores providing sea-ice extent data". | New line number(s): 174-175

We have updated the language to: "…and additional published cores providing sea ice extent data, SO136-111 and E27-23…" |
| RC1-28 | Figure 1 – please add abbreviations SSI and WSI in legend | Agreed – we will update Figure 1 accordingly. | New line number(s): 169

We have added a legend to Figure 1 that includes the abbreviations for SSI and WSI. |
| RC1-29 | sSST is sometimes written as SSST throughout the manuscript | Noted - we will standardize and update throughout the manuscript to SSST. | New line number(s): throughout

We have standardized sSST to SSST throughout the manuscript. |
| RC2-1 | From this paper alone, it is not clear what the percentage changes in SIC (%wSIC) represents. Does a value of 40% indicate that the amount of sea-ice is 40% of modern sea-ice concentrations or some other reference point? Or does it indicate that | Sea-ice concentration (SIC) is a pixel/grid-scale observation defined as the fraction of ocean area that is covered by sea ice. Sea-ice concentration thresholds are generally: 0-15% open ocean; 15-40% unconsolidated sea ice; >40% consolidated sea | New line number(s): 127-128; 331-333

We have added the following sentence to provide more clarity on what WSIC is measuring: "WSIC, which is a grid-scale observation of the mean state fraction of |

| | | | |
|---|---|---|---|
| | only 40% of the region around the core site is covered by sea-ice at this time? Furthermore, does %wSIC give any indication about what thickness of sea-ice is present? A couple of sentences in the methods section clarifying what "%wSIC" is would address this issue. | ice (Armand et al., 2005 and references therein; Hobbs et al., 216). Therefore, a value of 40% indicates that 40% of the region over the core site was covered by sea ice during the winter at the considered time slice. These values represent a mean state integrated over the time period covered by the sample.

As requested, we will provide additional clarity on what wSIC is measuring more specifically. | ocean area that is covered by sea ice over the sample period, and SSST…".

Lines 331-333 also provide reference to percentages for WSIC as follows:

"As outlined in Ferry et al., (2015), we consider <15% WSIC to represent an absence of winter sea ice, 15-40% WSIC as present but unconsolidated, and >40% to represent consolidated winter sea ice." |
| RC2-2 | Line 117-119: make it clear that these are modern(?) positions of sea ice extent and the subtropical/polar front. | Agreed – we will clarify that these are the modern positions of the sea-ice edge and fronts. | New line number(s): 179-180

We have added additional clarity to Figure 1. The caption now includes the following text:

"…red and blue lines show mean positions of modern summer sea ice (SSI) and winter sea ice (WSI) extents, respectively". |
| RC2-3 | Line 235: what proportion of the overall numbers of frustules counted in each sample are in the transfer function training set? If the | For TAN1302-96, the downcore proportion of diatoms included in the TF is >82% (mean = 92%). The Sea Ice group accounts for <1% during interglacials, and up to ~7% during glacials.

For SO136-111, the downcore proportion of diatoms included in the TF is >79% (mean = 91%). The Sea Ice | |

| | | | |
|---|---|---|---|
| | number (percentage) is low (<60%?) in any sample, are the sSST and %wSIC values compromised? | group accounts for <1% during interglacials, and up to ~4% during glacials.

No samples report using <60% of total identified specimens and therefore our SSST and WSIC estimates are not believed to have been compromised by low proportion of the TF diatom assemblages. | |
| RC2-4 | Line 308/309: is the Ferry et al (2015) data available for you to run through your transfer function? | The MAT has been applied on Ferry's data (core E27-23; Figure 1 included below). Results appear very similar to the published ones, especially in the timing of sea-ice changes. This was observed and published for core SO136-111 in Ferry et al., 2015. | |
| RC3-1 | Sedimentation rate in core TAN1302-96 is much higher during interglacial/warmer period than during glacial period and MIS 2, 3 and 4 are represented by less than 30cm in that core from ~90 to 120cm. The period that the authors discuss as MIS 3 is part of MIS 5. The evidence comes first from the d$^{18}$O stratigraphy measured on N. pachyderma (senestre? should be indicated by the authors). The values measured between 120 and 170cm are clearly too low to represent MIS 3. They indicate that from 120 to 300cm the sediments were deposited during MIS 5. This is also indicated by the 14C data: measurements at both | We appreciate the reviewer's deep engagement with the data provided in this manuscript. This comment has provided valuable discussion surrounding the robustness of the age model as currently outlined in the manuscript, and we welcome discussions to improve the reliability of our data and interpretations.

**Age model construction**

To test the reliability of our age model, we have constructed 4 additional age models (5 total) and have set up a series of tests to determine their reliability. All age models use the youngest 5 radiocarbon samples outlined in the | New line number(s): 182-222; Supplemental Online Materials (SOM)

We have provided additional age model construction and selection information in the SOM. This document includes the construction of 4 additional age models (including three that are tied to the EDT record, and one as suggested by Reviewer 3) and outlines our selection criteria.

The SOM compares: [1] the $\delta^{13}$C record of TAN1302-96 using all 5 age models with the $\delta^{13}$C record of SO136-111; [2] the calculated sedimentation rates for each age model compared with SO136-111; and [3] the $\delta^{18}$O for each age |

130 and 170cm indicated dates undistinguishable from background because both are older than 70kyr. Further evidence of the "extended" MIS 5 and shrinked MIS 2, 3, 4 could come from the carbon isotopic record of N. pachyderma but they are not presented in the paper. The authors could/should compare their isotopic record to the isotopic record of core SO136-111 that they are also using in this paper if they want further evidence. Correcting the chronology for the studied core TAN1302-96 will make it possible to reconcile the sea ice record of this core with those of the 2 cores from the same area: core E27-23 and core SO136-111. It is not clear if MIS 6 is represented in the core. There is no corresponding isotopic value but it might be due to the low resolution of the isotopic data

manuscript but use different tie points for the older portions of the core (incorporating comparisons with the EDC ($\delta$D as was done for SO136-111). The new age model versions also make different use of the 2 older, NDFB radiocarbon samples at 130 and 170cm. The 4 additional age models that we compare are as follows:

[1] **EDC 1**: includes both the 130cm and 170cm radiocarbon samples and is tied to the EDC SST record ($\delta$D data from Stenni et al. (2010) on the AICC2021 timescale from Veres et al. (2013)). The NDFB radiocarbon ages used were 57.5 ka and 57.7 ka, respectively, as these were the lower bracket of the NDFB results supplied by the CAS laboratory.

[2] **EDC 2**: includes only the 130cm radiocarbon sample (using an age of 57.5 ka) and excludes the 170cm sample. This model uses the same tie points to the EDC SST record as were used for EDC 1.

[3] **EDC 3**: excludes both the 130cm and 170cm radiocarbon samples and

model compared to the LR04 benthic stack.

We find that our original age model (with additional tie points and a slightly adjusted MIS 5) is the most robust of the constructed age models. We see the $\delta^{13}$C comparison (Figure S1 in the SOM) as particularly convincing evidence for the robustness of the d18O 1 age model.

Overall, this process has led to the refining of our original age model by increasing the number of tie points and more precisely tying them to the LR04 record.

uses the same tie points to the EDC SST record as were used for EDC 1.

[4] **R3**: based on the reviewer's comments, we attribute an age of 80 ka to the 130cm radiocarbon sample (reviewer suggests >70 ka) and a date of 104 ka to the 170cm sample. These date attributions are based on tying the TAN96 $\delta^{18}O$ record to the LR04 stack at the location of the radiocarbon samples. We then tie the TAN96 $\delta^{18}O$ record to the LR04 stack and assume that all sediment between 120 and 300cm accumulated during MIS 5, and that sediments between ~90 and 120 cm correspond to MIS 2, 3, and 4.

[5] **d18O 1**: original age model used in the manuscript, which uses 5 youngest radiocarbon dates, excludes the NDFB dates, and tie points between the TAN96 $\delta^{18}O$ record and the LR04 stack.

**Age model comparison:**

Following their selective tuning to EDC and $\delta18O$ records, we compare

| | | | |
|---|---|---|---|
| | | these 5 age models based on: | |
| | | [1] match of the $\delta^{13}C$ records between TAN 96 and nearby core SO136-111 | |
| | | [2] sensible behavior and magnitude of sediment accumulation rates relative to nearby SO136-111 | |
| | | [3] overall fit to $\delta18O$ and EDC SST | |
| | | **Observations** | |
| | | Our comparisons suggest: | |
| | | [1] Our current age model ($\delta18O$ 1) fits very well with the $\delta13C$ record from SO136-111 (Figure 2), suggesting that it is consistent with the age model provided for SO136-111. | |
| | | [2] R3, EDC 3, and d18O 1 all provide reasonable fits to the LR04 stack | |
| | | [3] $\delta18O$-1 and EDC 3 provide the most sensible sedimentation rates when compared with SO136-111 rates. | |
| | | **Additional thoughts:** | |
| | | Overall, we find it unlikely that the TAN96 sedimentation rates would be reduced to | |

near-zero (~0.2 cm/ka) during glacial periods (i.e., ~13cm (117 to 130cm) deposited over >60 ka for MIS 2, 3, & 4), as was suggested by Reviewer 3.  While it makes sense that polar cores with >80% sea ice cover would experience greatly reduced sedimentation rates, the wSIC estimates for TAN 96 are 40-50% during glacial periods, suggesting that the study area would experience some productivity during glacial times. Furthermore, the proximal site SO136-111- which has comparable wSIC estimates during glacial periods - does not exhibit this behavior, as sedimentation rates are between 2-3 cm/ka during glacial periods.

**Conclusion:**

Our final response to reviewer document will provide the details of these comparisons, and our revised manuscript will provide a more comprehensive explanation of our age model determination including all supporting information.

| RC3-2 | Chronology: As the authors indicate , significant MRA variability occurs over a glacial cycle, specifically in the southern high latitudes. They should use as a minimum ±100 years for the uncertainty on the MRA as it is the variation indicated by Paterne et al., 2009, for the last century. The authors do not indicate the uncertainty they evaluate for the tie points used to correlate the planktic isotopic record to the LR04 benthic record. From figure 3 it seems that they also choose a too small uncertainty. Anyway the authors should give more details. Furthermore as they present a planktic isotopic record and a SST record and as their goal is to discuss the impact of sea ice extent on atmospheric $CO_2$, it would make more sense to establish the chronology of MIS 5 comparing their records to EDC deuterium record, following Govin et al., 2015, Capron et al., 2014. Anyway the record resolution is pretty low (partly due to the low sedimentation | We have taken this comment into consideration and are in the process of comparing alternative age models that use the EDC deuterium record.

As noted above, we will also provide more information on the age model that is selected, including a minimum ±100 year uncertainty for the MRA and more information on tie point uncertainty. | New line number(s): 199-202

We have added an additional sentence outlining the uncertainty associated with the LR04 stack and tie points (± 4 ka) and have also updated the MRA variability to ± 100 years, in line with Paterne et al. (2009).

In line with the above comment (RC3-1), we have constructed 3 additional age models using the EDT record; however, as outlined above, we have determined that the d18O 1 age model (with additional tie points) is the most robust produced. |

| | | | |
|---|---|---|---|
| | rate of the core) so the real uncertainties are large and this comment is not that important. | | |
| RC3-3 | SST and wSIC: the authors should give more details: how many analogues have been used for reconstructions? Is the error indicated on the figure the standard deviation between the different analogues? The tables should be available to reviewers. | In this version, the MAT has a ~1°C RMSEP for SSST and a 10% RMSEP for WSIC on the modern validation step. These errors are generally applied downcore as other TF (IKM, WA-PLS…) and geochemical proxies (TEX86, UK37, Mg/Ca) only provide a mean error on the calibration. However, MAT allows for a sample-only error, calculated as the standard deviation of the chosen analogs. In core TAN96, SSST standard deviation varies between ~0.2°C and ~2°C (mean of 1.28°C, in good agreement with the modern calibration) (Figure 3). WSIC standard deviation varies between 0% during interglacials to ~30% in glacials (mean of 8%). | New line number(s): 389-390; 434-436 |
| | | | In addition to the information previously provided, we have added the following sentences: |
| | | | "There were no non-analog conditions observed in TAN1302-96 samples and all estimates were calculated on five analogs." |
| | | | And |
| | | | "In core SO136-111, the 33 species included in the transfer function represent values >79% of the total diatom assemblages (mean of 91%). There were no non-analog conditions observed in SO136-111 samples and all estimates were calculated on five analogs" |
| | | In SO136-111, SSST standard deviation varies between ~0.5°C and ~2°C (mean of 1.14°C) (Figure 3). WSIC standard deviation varies between 0% in interglacials to ~25% during glacials (mean of 9.96%). All | As requested, we are in the process of updating the online dataset on Pangae to include the requested information, including additional data on the analogs, thresholds used, dissimilarity |

|  |  | these data are provided in the appendix table. | coefficients, and other data. |
|---|---|---|---|
|  |  | In TAN96, the dissimilarity of the fifth analog varies between ~0.15 and ~0.4 (mean of 0.23), far below the threshold of the first quartile (0.7). All five analogs are always preserved and estimates & SD are done on 5 analogs. |  |
|  |  | In core SO136-111, the dissimilarity of the fifth analog varies between ~0.05 and ~0.3 (mean of 0.14), far below the threshold of the first quartile (0.7). All five analogs are always preserved and estimates & SD are done on 5 analogs. |  |
|  |  | These data are not pivotal to the manuscript but a mention to the good dissimilarity and to the calculation on 5 analogs will be added to the revised manuscript. |  |
| RC3-4 | Results: what is indicated in the text is not what is presented on the figures. Some examples: line 253, the SST increase seems to be ≤3°C on the figure. Taking into account uncertainties ~1 to 4°C would be precise | This is a good catch – after looking through the datafile, an additional blank line of data was accidently added causing some of the data points to be shifted and/or not included in the figure. | New line number(s): 381; 394-395

We have corrected Figure 4 to include the proper data (see initial response for more information). |

| | enough. Line 254: on the figure the 2 methods indicate ~22 to ~33% wSIC for the oldest point. Where does the 48% comes from? Line 256: I do not see a rise in SST during MIS 5e, only variability. | An updated figure is provided below (Figure 4) which shows the wSIC of 48% at 140 ka. We note that the SSST value at the MIS 5e/6 boundary is 4 °C in the updated figure, and the text will be updated to reflect the corrected value.

Finally, we do not disagree with your observations regarding SSTs during MIS 5e. We will change the wording to something like:

"Reconstructed SSSTs were variable throughout MIS 5e, reaching a maximum..." | We have also updated line 272-273 to read:

"Reconstructed SSST were variable throughout MIS 5e..."

We have carefully reread the manuscript to identify and update similar errors. |